# Expert Proximity as Surrogate Rewards for Single Demonstration Imitation Learning

## Abstract

This study investigates the challenging single-demonstration imitation learning (IL) setting. In this context, the learning agent relies solely on a single expert demonstration and operates in an environment that lacks external reward signals, human feedback, or prior analogous knowledge, as obtaining multiple demonstrations or engineering complex reward functions is often infeasible. Given these constraints, the study introduces a methodology termed Transition Discriminator-based IL (TDIL). TDIL aims to augment the density of available reward signals and enhance agent performance by incorporating environmental dynamics. It posits that rather than strictly adhering to a limited expert demonstration, the agent should first aim to reach states proximal to expert behavior. The study introduces a surrogate reward function, approximated by a transition discriminator, to facilitate this process. TDIL demonstrates promise in addressing the sparse-reward problem common in single-demonstration IL, and stabilizing the learning process of the agent during training. A comprehensive set of experiments across multiple benchmarks validate the effectiveness of TDIL over existing IL methods.

## 1 Introduction

This study aims to explore a specialized but pragmatically significant paradigm within the realm of imitation learning (IL), referred to as *single-demonstration IL*. Under this paradigm, the learner is limited to a solitary expert demonstration for reference while being permitted to interact with the environment. However, the learner is not granted access to environmental reward signals, online human feedback, or prior knowledge acquired from analogous tasks. These constraints make this IL paradigm particularly pertinent to real-world endeavors, such as the training of autonomous robots, primarily because the acquisition of multiple demonstrations or the meticulous construction of reward functions for reinforcement learning (RL) agents in practical scenarios could be not only strenuous but also expensive. In light of these limitations, developing an effective and robust learning mechanism capable of operating solely with a single demonstration becomes essential and advantageous. It mitigates the necessity for extensive demonstrations, continuous human supervision, and intricate reward engineering. Although this research domain bears considerable significance, it remains relatively unexplored, and leaves room for the advancement of contemporary IL approaches.

The most formidable obstacle in the *single-demonstration IL* paradigm lies in the limited availability of expert demonstrations. This scarcity often results in a sparse reward system for many inverse reinforcement learning (IRL) methods, as elaborated in Section 4 and Appendix A.1. These methods typically allocate rewards solely when an agent reaches an expert state and executes an expert action. Such sparse reward systems could create hurdles for the agents during exploration when they operates outside the scope of available expert data, which in turn diminishes the effectiveness of these derived signals. The issue of reward sparsity becomes even more pronounced in high-dimensional, continuous environments with randomly initialized positions. In such settings, an agent with limited understanding of environmental dynamics may face difficulty in reaching expert states, and results in receiving zero training signal. For other IL methods like behavior cloning (BC) (Bain & Sammut, 1995), the absence of expert demonstrations can lead to overfitting. In addition, the limited expert support increases the likelihood of the agent encountering unfamiliar states and performing suboptimally. Therefore, in *single-demonstration IL*, devising a method that adapts to the limited expert data, addresses sparse reward signals, and achieves expert-level performance presents a challenge.

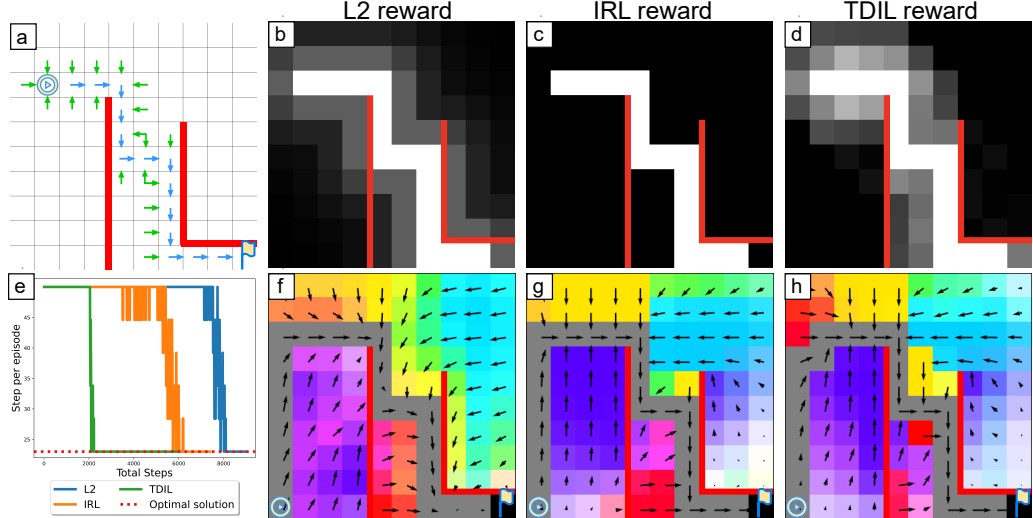

Figure 1: A motivational grid-world example for comparing different IL methods trained with a single expert demonstration. (a) depicts the expert's demonstration, denoted by blue arrows, while red lines represent impassable barriers, reflecting environmental dynamics. The green arrows symbolize the state-action pairs that are one step directed toward the expert states. Subfigures (b)-(d) present reward signals calculated through various methods: (b) based on L2 distance between the agent's and the expert's state-action pairs, (c) using an optimal discriminator $p(\mathcal{O}_t|s_t, a_t)$ as reward (Levine, 2018), and (d) through our proposed transition discriminator. Subfigure (e) shows the total steps needed for each agent, trained by different reward schemes, to reach the goal. Finally, subfigures (f)-(h) illustrate the actions calculated by averaging the direction represented by the logits for the discrete actions from the learned policy at distinct grid locations

To confront the challenges in single-demonstration IL, this study introduces a methodology termed Transition Discriminator-based IL (TDIL). TDIL aims to increase the density of obtainable reward signals in the IRL setting while accounting for environmental dynamics to ensure robust agent performance. A motivational example illustrating this concept is provided in Fig. 1 (a). As a single demonstration may be insufficient to fully encapsulate the expert policy, TDIL adopts a more relaxed criterion. In contrast to conventional methods that only compel the optimal policy to adhere strictly to expert demonstrations, as in Behavioral Cloning (BC), TDIL posits that an optimal policy should first transition toward the expert support before conforming to the solely available expert demonstration. If an agent finds itself in a state that allows for a direct transition to an expert state (e.g., the green arrows in Fig. 1 (a)), the optimal action is to facilitate this transition. In other words, our approach is predicated on the assumption that the agent, when not in an expert state, should opt for immediate actions leading to expert's proximity states whenever possible. In practice, given that multiple transitions may be feasible in these non-expert state-action pairs, TDIL introduces a surrogate reward function that rewards the agent for moving toward states in close proximity to expert states. We demonstrate that this is equivalent to taking the optimal action at those proximity states in Section 3. This adjustment enhances the density of the reward signal distribution and addresses the inherent challenge of sparse rewards during the training phase. As the surrogate reward function is not directly computable, this study further introduces a *transition discriminator* for approximating surrogate rewards. The transition discriminator adopts a training objective aimed at distinguishing between valid and non-valid transitions regarding two states' reachability of a given environment. Instead of conventional discriminators trained by the min-max strategy, the transition discriminator can be trained stably using the interactions that the agent collects in the environment during training.

To validate the efficacy of the proposed methodology, we perform comprehensive experiments on five widely adopted MuJoCo (Todorov et al., 2012) benchmarks, aligning with most prior IL research, as well as the "Adroit Door" (Rajeswaran et al., 2017) environment in the Gymnasium-Robotics collection (de Lazcano et al., 2023). The experimental evidence reveals that our methodology delivers exceptional performance, matches expert-level results on these benchmarks, and outperforms existing IL approaches. Another key insight from our experiments is a significant correlation

between the derived reward signals and the inaccessible ground truth reward signals. This correlation offers a practical solution for blind model selection, and effectively removes the dependency on external signals such as rewards. This differentiates TDIL from prior work that relied on environment rewards at test-time for early termination or optimal model selection, which are assumptions impractical in the general IL context. The contributions of this study can be summarized as follows:

1. We highlight the challenges of single-demonstration IL characterized by scarce expert data, a condition often encountered in practical scenarios that may result in sparse reward signals.

2. We utilize surrogate rewards, approximated by a transition discriminator, to capture both proximity to a single, sparse expert demonstration and awareness of environmental dynamics, with the aim of enhancing convergence speed in the single-demonstration IL settings.

3. We introduce TDIL, a practical and non-adversarial approach using a transition discriminator to implement the surrogate reward, with the aim of maximizing its cumulative sum.

4. We unveil a practical approach for blind model selection by disclosing a significant correlation between derived reward signals and inaccessible ground truth rewards.

5. We provide empirical proof to demonstrate that TDIL outperforms existing IL approaches in multiple benchmarks, showcasing its robustness with only a single expert demonstration.

## 2 PRELIMINARY

**Reinforcement learning (RL).** An MDP is typically formalized as a tuple $\langle \mathcal{S}, \mathcal{A}, P, R, p_0 \rangle$, where $\mathcal{S}$ represents the state space, $\mathcal{A}$ the action space, $P : \mathcal{S} \times \mathcal{A} \times \mathcal{S} \to \mathbb{R}$ the transition function, $R(s, a) : \mathcal{S} \times \mathcal{A} \to \mathbb{R}$ the reward function, and $p_0(s_0)$ the distribution of the initial state $s_0$. The transition function $P(s_{t+1}|s_t, a_t)$ specifies the probability of transitioning to state $s_{t+1}$ upon taking action $a_t$ in state $s_t$. Within this MDP, a trajectory $\tau$ is defined as a sequence of states and actions $[s_0, a_0, s_1, a_1, \ldots, s_T, a_T]$, where $s_0$ is sampled from the distribution $p_0$, and $s_{t+1}$ is the resulting state after taking action $a_t$ in state $s_t$. The objective of an RL policy $\pi(a|s, \theta)$ is to learn a set of parameters $\theta$ that maximizes the expected total return $\mathbb{E}_{\tau \sim p(s,a|\theta)} \sum_{t=0}^{T} R(s_t, a_t)$. The probability $p(\tau)$ of sampling a given trajectory $\tau$ can be expressed as $p_0(s_0) \prod_{t=0}^{T} \pi(a_t|s_t, \theta) P(s_{t+1}|s_t, a_t)$.

**Single-demonstration imitation learning (single-demo IL).** During the training of single-demo IL settings, the agent can interact with the environment. However, it does not have access to the reward function $R$. The agent is also given a trajectory $\tau_e = [s_0^e, a_0^e, s_1^e, a_1^e, \ldots, s_T^e, a_T^e]$ generated by an expert policy $\pi_e$ in the same environment as a hint of the reward $R$. Hence, the goal of single-demo IL is to train a policy that can converge to the expert demonstration even when initiated from a different initial state $s_0$, and faithfully follow the expert's actions when within the support of the demonstration. After training, the performance is evaluated by the ground truth reward function $R$.

**Inverse reinforcement learning (IRL).** IRL methods constitute a type of IL that aims to learn or infer the reward function based on provided demonstrations. Levine (2018) demonstrated that the objective of IRL is to learn a Conditional Probability Distribution (CPD) denoted as $p(\mathcal{O}_t = 1|s_t, a_t)$. In this expression, the optimal indicator $\mathcal{O}_t$ serves as a binary random variable that indicates whether the time step $t$ is optimal. Specifically, in the context of IRL, $\mathcal{O}_t = 1$ if the $(s_t, a_t)$ pair is present in an expert trajectory. Furthermore, the CPD $p(\mathcal{O}_t = 1|s_t, a_t)$ can be marginalized to form $p(\mathcal{O}_t = 1|s_t) = \int_{\mathcal{A}} \pi(a_t|s_t) p(\mathcal{O}_t = 1|s_t, a_t) da_t$. By assuming the policy $\pi(a|s)$ produces the expert's actions in the expert's states, $p(\mathcal{O}_t = 1|s_t) = 1$ if and only if $s_t$ is an expert state. The assumption can be ensured through BC or GAIL.

## 3 METHODOLOGY

In this section, we start with a motivational example in Section 3.1 to showcase the limitations of existing IL methods and the benefits of TDIL in the single-demonstration setting. In Section 3.2, we define our surrogate reward and propose using a transition discriminator to approximate it. Finally, in Section 3.3, we propose our TDIL algorithm to train both the agent and the transition discriminator.

### 3.1 MOTIVATIONAL EXAMPLE : IL UNDER SINGLE DEMONSTRATION CONSTRAINT

Most IRL algorithms try to match the distribution of a learning agent (Dadashi et al., 2021) with the distribution of the given expert. However, in a high dimensional continuous environment, a single demonstration is hard to be matched by the agent support since the states in the demonstration are just a 1-manifold (a dimension line) in the state space, which is disjoint from the agent support (Arjovsky & Bottou, 2017). This makes the traditional IRL methods (referred to as IRL in this paper) that minimize a f-divergence (Ho & Ermon, 2016) even more unstable in single-demo IL settings.

Our intuition for solving this problem is assuming the state-action pairs are also optimal if they can reach an expert demonstration state. Take the 2D grid-world toy example in Fig. 1 (a) as an example (rules are in the caption). The blue arrows are the given demonstration, and the green arrows are the state-action pairs that we assume are optimal. Although the green arrows are not the shortest path to the destination (the flag), going back to the states of the demonstration is the safest decision on out-of-distribution states since we do not have access to the ground truth reward function. With our assumption, a 1-manifold demonstration can be densified to a support. This can be illustrated in the toy example where Fig. 1 (c) is the traditional IRL reward, and Fig. 1 (d) is our reward system. With our denser reward function, an SAC (Haarnoja et al., 2018) agent can learn a policy (Fig. 1 (g) and (h)) of the whole map three times faster than traditional IRL which is shown in Fig. 1 (e). Moreover, the density difference between our reward system and traditional IRL rewards is even larger in a high-dimensional continuous environment. It is almost impossible for the agent to reach any expert state during exploration, even if the agent starts from an initial state close to the single demonstration.

Similar to our intuition, some of the IRL methods like PWIL (Dadashi et al., 2021), FISH (Haldar et al., 2023b), and ROT (Haldar et al., 2023a) also provide rewards when the agent is "close" to the demonstration. However, the distance measuring functions they used, like Euclidean and Cosine distance, are not theoretically guaranteed. For instance, in the toy example (Fig. 1 (a)), although some grid cells are adjacent to each other, they are not "close" since there is a barrier between them (red lines). Take self-driving as another example, since the car can only move forward or backward, the positions on the side are also not "close". To demonstrate the issue of using a distance measuring function that does not consider the dynamic of the environment, we conduct the same SAC experiment with L2 reward (Fig. 1 (b)) in the toy example. The results (Fig. 1 (e)) show that it is even worse than the traditional IRL reward since the agent will be stocked in some states that are actually not "close" to expert states. In addition, besides using a distance measuring function that considers the dynamic, our method is fundamentally different from those IRL methods. Our main idea is to propose a new and denser distribution from a single demonstration for the agent to match. This means all the matching methods, such as the Wasserstein distance used in PWIL, can be used to improve our method.

### 3.2 EXPERT PROXIMITY AND SURROGATE REWARDS

In this section, we first introduce the expert reachability (ER) indicator $\tilde{\mathcal{O}}_t$ to formally define the green arrows in Fig. 1 (a). Next, we use the ER indicator to define expert proximity, which later is used in our surrogate reward function $R_{\text{TDIL}}(s_t, a_t)$. Finally, we show that $R_{\text{TDIL}}(s_t, a_t)$ can be approximated by transition dynamic.

**Define expert proximity and surrogate reward.** Given a state-action pair $(s_t, a_t)$, we use an expert reachability indicator $p(\tilde{\mathcal{O}}_t = 1|s_t, a_t)$ to determine the probability of reaching an expert state by selecting action $a_t$ in state $s_t$. Take Fig. 1 (a) as an example, if $(s_t, a_t)$ is one of the green arrows, $p(\tilde{\mathcal{O}}_t = 1|s_t, a_t) = 1$ since the agent can reach an expert state from state $s_t$. Formally, we define $p(\tilde{\mathcal{O}}_t = 1|s_t, a_t)$ as:

$$p(\tilde{\mathcal{O}}_t = 1|s_t, a_t) \stackrel{\text{def}}{=} \int_{\mathcal{S}} P(s_{t+1}|s_t, a_t)p(\mathcal{O}_{t+1} = 1|s_{t+1})ds_{t+1}, \tag{1}$$

where $P$ is the transition function and $p(\mathcal{O}_{t+1} = 1|s_{t+1})$ is the probability of $s_t$ is an expert state.

Theoretically, $p(\tilde{\mathcal{O}}_t = 1|s_t, a_t)$ can be used as the reward system if we have access to the transition function (e.g., Fig. 1 (a)). However, the transition function is hard to access in most of the environ-

ments. In addition, since RL agents are forced to explore nonoptimal actions during training, using $p(\tilde{\mathcal{O}}_t = 1|s_t, a_t)$ as a reward can still result in a sparse reward system in a high dimensional continuous environment. Hence, we further define the concept of *expert proximity*: Expert proximity is a state set. All the states in expert proximity are one action reachable to the expert states. In other words, a state $s_t$ is in expert proximity if and only if $\tilde{\mathcal{O}}_t = 1$. Given a policy $\pi(a|s)$, $p(\tilde{\mathcal{O}}_t = 1|s_t)$ can be derived from $p(\tilde{\mathcal{O}}_t = 1|s_t, a_t)$ as following:

$$
\begin{aligned}
p(\tilde{\mathcal{O}}_t = 1|s_t) &= \int_{\mathcal{A}} \pi(a_t|s_t)p(\tilde{\mathcal{O}}_t = 1|s_t, a_t)da_t \\
&= \int_{\mathcal{A}} \pi(a_t|s_t) \int_{\mathcal{S}} p(s_{t+1}|s_t, a_t)p(\mathcal{O}_{t+1} = 1|s_{t+1})ds_{t+1}da_t \\
&= \int_{\mathcal{S}} p(\mathcal{O}_{t+1} = 1|s_{t+1}) \int_{\mathcal{A}} \pi(a_t|s_t)p(s_{t+1}|s_t, a_t)da_t ds_{t+1}.
\end{aligned}
\tag{2}
$$

Notice that the $p(\tilde{\mathcal{O}}_t = 1|s_t)$ is derived with the ground truth expert support. However, since we do not have access to expert support, we can only calculate the probability $\hat{p}(\tilde{\mathcal{O}}_t = 1|s_t)$ of reaching the given trajectory $\tau$:

$$
\begin{aligned}
\hat{p}(\tilde{\mathcal{O}}_t = 1|s_t) &= \sum_{i=0}^{T} p(\mathcal{O}_{t+1} = 1|s_i^e) \int_{\mathcal{A}} \pi(a_t|s_t)p(s_i^e|s_t, a_t)da_t \\
&= \sum_{i=0}^{T} \int_{\mathcal{A}} \pi(a_t|s_t)p(s_i^e|s_t, a_t)da_t,
\end{aligned}
\tag{3}
$$

where $s_i^e$ is the $i$th state in the expert trajectory $\tau$ and $T$ is the total timesteps of $\tau$. Finally, we define our surrogate reward function $R_{\text{TDIL}}(s_t, a_t)$ as:

$$
R_{\text{TDIL}}(s_t, a_t) \stackrel{\text{def}}{=} \mathbb{E}_{s_{t+1} \sim p(s_{t+1}|s_t, a_t)}\left[\hat{p}(\tilde{\mathcal{O}}_{t+1} = 1|s_{t+1})\right].
\tag{4}
$$

Surrogate reward function $R_{\text{TDIL}}(s_t, a_t)$ encourages the agent to navigate to states that are in expert proximity. This results in a denser (higher dimension) reward when compared to the IRL reward function $R_{\text{IRL}}$, which only assigns rewards to state-action pairs that follow the expert trajectory as discussed in Section 3.1. However, $R_{\text{TDIL}}(s_t, a_t)$ does not encourage the agent to follow the expert's action. Hence, to ensure optimality on the expert's states, the agent is trained with the aggregate reward $R_{\text{agg}}$, which is defined as:

$$
R_{\text{agg}}(s_t, a_t) \stackrel{\text{def}}{=} \beta R_{\text{IRL}}(s_t, a_t) + (1 - \beta)R_{\text{TDIL}}(s_t, a_t),
\tag{5}
$$

where $\beta$ is a hyperparameter used to balance two rewards.

**Approximate the surrogate reward with transition discriminator.** To compute $R_{\text{TDIL}}$, we need to be able to calculate $\int_{\mathcal{A}} \pi(a_t|s_t)P(s_i^e|s_t, a_t)da_t$. By assuming $\pi(a_t|s_t)$ is optimal, and $s_t$ can only reach at most one expert state, $\int_{\mathcal{A}} \pi(a_t|s_t)P(s_i^e|s_t, a_t)da_t$ can be converted to:

$$
\max_{a_t} P(s_i^e|s_t, a_t).
\tag{6}
$$

Eq. 6 is identical to the capability of moving from $s_t$ to $s_i^e$ and can be solved if we have the reachability probability $P(s_i, s_j) = \max_a P(s_j|s_i, a)$. However, we do not have the proper data to train a reachability function during training since the tuples in the replay buffer are in a format of $(s_t, a_t, s_{t+1})$. Hence, instead of predicting the reachability probability, we train a transition discriminator $D_\phi(s_i, s_j)$, which can determine whether a given state $s_i$ can transit to another state $s_j$ within a single time step. For example, for any tuple $(s_t, a_t, s_{t+1})$ in the replay buffer, $D_\phi(s_t, s_{t+1})$ should return 1 since the tuple evidences the reachability. Formally, the transition discriminator $D_\phi(s_i, s_j)$ is defined as:

$$
D_\phi(s_i, s_j) \stackrel{\text{def}}{=} \max_{a_i} \mathbb{1}[P(s_j|s_i, a_i) > 0].
\tag{7}
$$

and the surrogate reward function $R_{\text{TDIL}}$ becomes:

$$
R_{\text{TDIL}}(s_i, s_j) \approx \mathbb{E}_{s_{t+1} \sim p(s_{t+1}|s_t, a_t)}\left[\sum_{i=0}^{T} D_\phi(s_{t+1}, s_i^e)\right].
\tag{8}
$$

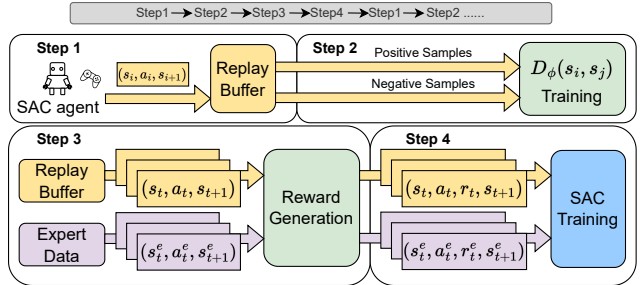 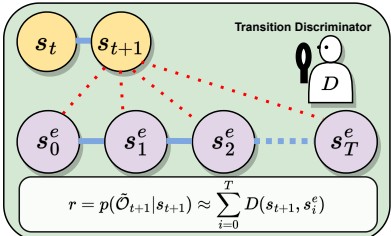

Figure 2: Overview of TDIL. **Step 1:** Agent-environment interaction. **Step 2:** Transition discriminator updates. **Step 3:** Generation of surrogate rewards. **Step 4:** Agent updates.

Figure 3: A detailed illustration of the reward generation block and how the transition discriminator operates.

Note that the reward will later be used by some actor-critic algorithms as the normal reward, and the expectation in Eq. 8 can be safely ignored. With actor-critic algorithms, the reward signal will be propagated to the earlier states that are not in expert proximity through the TD updates of the value function, which can then used to guide the policy from multi-steps away from the expert's distribution.

### 3.3    TDIL Training Algorithm

In this section, we first introduce our training framework: Transition Discriminator-based IL (TDIL). Next, we talk about the details of how to train a transition discriminator. Finally, we talk about blind model selection under TDIL.

**Overview of TDIL framework**   Fig. 2 presents an overview of the TDIL framework, which involves several key steps. First, the agent interacts with the environment to collect transitions. These transitions are then utilized to update the transition discriminator in the subsequent step. The third step involves sampling a batch of collected transitions and calculating the reward for each transition using the transition discriminator, as illustrated in Fig. 3. In a similar fashion, the reward for the expert data is also calculated using the transition discriminator. Please note that these are not the ground truth rewards for the expert data but the rewards inferred by the transition discriminator. In the fourth step, both the agent's and expert's transitions are utilized for training and updating the actor and critic of the Soft Actor-Critic (SAC) algorithm (Haarnoja et al., 2018). Note that, in the main experiments we do not use the $R_{\text{IRL}}$ ($\beta = 0$), we use the BC loss to ensure the policy $\pi$ is optimal on the expert states (Section 2) instead. Note that the pseudo code and more details of TDIL algorithm can be found in Section A.2. The ablation study on different $\beta$ value selection can be found in Section A.4.8

**Training of transition discriminator.**   For the transition discriminator $D_\phi$, we optimize it with maximum likelihood training. The binary cross-entropy loss $L_D(\phi)$ is as follows:

$$L_D = -\Big(\alpha \mathbb{E}_{(s_i, s_j) \sim B^+} \big[\log\big(D(s_i, s_j)\big)\big] + (1-\alpha)\mathbb{E}_{(s_i, s_j) \sim B^-} \big[\log\big(1 - D(s_i, s_j)\big)\big]\Big), \quad (9)$$

where $\alpha \in (0, 1)$ is a balancing coefficient, $B^+$ is the set of positive samples, and $B^-$ is the set of negative samples. In practice, we choose the set of positive samples $B^+$ as:

$$B^+ = \{(s, s') \mid (s, a, s') \in B\}, \quad (10)$$

where $B$ is the replay buffer. For negative samples $B^-$, we choose the union of the set of contrastive samples (i.e., easy negative samples) $B^-_{\text{contrastive}}$ and the set of reversed transition samples (i.e., hard negative samples) $B^-_{\text{reversed}}$ as:

$$\begin{aligned} B^- &= B^-_{\text{contrastive}} \cup B^-_{\text{reversed}}, \text{ where} \\ B^-_{\text{contrastive}} &= \{(s_i, s_j) \mid (s_i, a_i, s_{i+1}), (s_j, a_j, s_{j+1}) \in B\}, \text{ and} \\ B^-_{\text{reversed}} &= \{(s', s) \mid (s, a, s') \in B\}. \end{aligned} \quad (11)$$

The intuition here is to take the valid transition states collected by the agent as positive samples. On the other hand, for the negative samples, we assume randomly sampled two states has a really low

chance of being a valid transition, and most reversed transition are invalid transition. This training process utilizes transition data collected from agent-environment interactions in a self-supervised manner. Unlike conventional supervised learning, the proposed methodology does not require any human-expert labeling and is entirely data-driven.

**Blind Model Selection** Unlike many IRL rewards, our reward function can select a decent model from all the training checkpoints after normalization without the help of ground truth reward $R$. This is an important practical problem since most IL algorithms' best model is not the model that trained for the longest (see Appendix A.4.5 ). To handle this problem, instead of using the raw return $\sum_{t=0}^{\tilde{T}} R_{\text{TDIL}}(s_t, a_t)$ we use a normalized relative return $\sum_{t=0}^{\tilde{T}} R_{\text{TDIL}}(s_t, a_t) / \sum_{t=0}^{T} R_{\text{TDIL}}(s_t^e, a_t^e)$ to select the best model. This is because the transition discriminator $D_\phi$ differs in different checkpoints. See Appendix A.4.4 for more implementation details.

## 4 RELATED WORK

Existing IL approaches exhibit limitations in the single-demonstration IL setting. For example, in *one-shot IL* (Duan et al., 2017; Finn et al., 2017; Dasari & Gupta, 2021; Yu et al., 2018; Mandi et al., 2022; Huang et al., 2019; Netanyahu et al., 2022; Valassakis et al., 2022; Hu et al., 2020; Sontakke et al., 2023), researchers have explored the use of meta-demonstrations, which are demonstrations associated with other tasks, as a tool for pre-training before proceeding to one-shot adaptation. Nevertheless, gathering a substantial volume of meta-demonstrations, which are necessary for training meta parameters prior to their one-shot utilization, could be infeasible due to the expensive nature of expert demonstrations. In addition to one-shot IL, *online IL* methods also encounter challenges in the single-demonstration IL contexts due to two primary reasons. First, distribution matching methods with a min-max formulation (Ho & Ermon, 2016; Fu et al., 2018; Ke et al., 2021; Ghasemipour et al., 2020; Ni et al., 2020; Swamy et al., 2021; Kostrikov et al., 2020; Camacho et al., 2021; Freund et al., 2023; K. et al., 2019; Han et al., 2022; Zeng et al., 2022; Viano et al., 2022) might induce potential instability and sub-optimality in sparse demonstration data situations, which can compromise the effectiveness and reliability of these methods. Second, methods that rely on expert support estimation (Wang et al., 2019; Brantley et al., 2020; Liu et al., 2020; Kim et al., 2020) often face difficulties when expert data or demonstrations are limited. This is attributable to their reliance on the availability and quality of expert demonstrations, which leaves them ill-suited for scenarios with scarce expert data. Other IL approaches (Dadashi et al., 2021; Xiao et al., 2019), on the other hand, are also less suitable for the single-demonstration IL setting, as they tend to overlook environmental dynamics. Specifically, these approaches might identify certain states as being close or similar based on their proximity in Euclidean space, even though these states may not be permissible for transition in a Markov Decision Process (MDP). This limitation impairs their capacity to capture the complexity and variability of environments. Moreover, the majority of the aforementioned methods either struggle to achieve expert-level performance in high-dimensional environments, or are less adept at achieving robust generalization (Ni et al., 2020; Freund et al., 2023; Dadashi et al., 2021; Al-Hafez et al., 2023). These constraints highlight the necessity for enhanced strategies in the single-demonstration IL setting. The key objectives include accommodating limited expert data, while taking into account environmental dynamics. Please refer to Appendix A.1 for more details.

## 5 EXPERIMENTAL RESULTS

This section presents our experimental results on MuJoCo environments, ablation studies, discuss the insights, and compares our TDIL against the baselines.

### 5.1 EXPERIMENTAL SETUP

**Environments.** We evaluate the performance of our method and the baselines on a number of MuJoCo environments, including *HalfCheetah-v3*, *Hopper-v3*, *Ant-v3*, *Humanoid-v3*, and *Walker2d-v3*. The experiments in Adroid Hand environment are shown in Appendix A.4.3

**Expert demonstration.** We use SAC (Haarnoja et al., 2018) to generate expert demonstrations for different environments. The SAC experts are trained using the default parameters till convergence.

Table 1: Performance evaluation results of different methods results with oracle model selection.

| | BC (Pomerleau, 1991) | GAIL (Ho & Ermon, 2016) | f-IRL (Ni et al., 2020) | PWIL (Dadashi et al., 2021) | CFIL (Freund et al., 2023) | Ours | Expert |
|---|---|---|---|---|---|---|---|
| HalfCheetah-v3 | $211 \pm 49$ | $693 \pm 158$ | $14{,}560 \pm 823$ | $11{,}460 \pm 4{,}774$ | $13{,}636 \pm 1{,}695$ | $\mathbf{15{,}666} \pm 85$ | 15,251 |
| Hopper-v3 | $507 \pm 161$ | $3{,}209 \pm 372$ | $3{,}693 \pm 162$ | $3{,}849 \pm 209$ | $\mathbf{4{,}131} \pm 34$ | $4{,}115 \pm 14$ | 4,114 |
| Ant-v3 | $990 \pm 6$ | $966 \pm 22$ | $5{,}597 \pm 194$ | $5{,}579 \pm 314$ | $5{,}812 \pm 2{,}692$ | $\mathbf{6{,}434} \pm 66$ | 6,561 |
| Humanoid-v3 | $429 \pm 20$ | $582 \pm 52$ | N/A | $5{,}499 \pm 94$ | $5{,}354 \pm 337$ | $\mathbf{5{,}758} \pm 173$ | 5,855 |
| Walker2d-v3 | $299 \pm 75$ | $554 \pm 217$ | $4{,}746 \pm 316$ | $3{,}391 \pm 1{,}873$ | $2{,}402 \pm 949$ | $\mathbf{6312} \pm 47$ | 6,123 |

**Baselines.** We selected BC, GAIL, f-IRL, PWIL, and CFIL as our baselines. BC serves as the lower bound, representing the minimal performance expected from online IL approaches. GAIL is a widely recognized and extensively adopted method in IL. CFIL and f-IRL both represent state-of-the-art adversarial based methods, while PWIL serves as a representative of non-adversarial approaches. Note that IQ-learn is excluded from our comparison due to its subpar performance, a conclusion supported by several recent studies (Zeng et al., 2022; Sikchi et al., 2022). Moreover, it should also be noted that the original f-IRL paper does not include experiments on *Humanoid-v3*. We endeavored to faithfully implement their code on the Humanoid environment, but the performance fell short of expectations. We omitted LS-IQ from our study due to the absence of its complete code.

**Hyperparameters.** For all the baselines, we utilize their open-source implementation except for PWIL. The reason is that PWIL was originally implemented using the D4PG algorithm, a distributional algorithm unsuitable for our context. To ensure a fair comparison, we re-implemented PWIL on the SAC backbone and fine-tuned it to achieve its optimal performance. To maintain consistency, all algorithms are run for 3M timesteps, each with five random seeds. We evaluate the performance over ten rollouts for different random seeds. To validate the performance of our reward $R_{\text{TDIL}}$ and reduce the training cost, we set the value of $\beta$ in Eq. (5) to 0. The details are offered in Appendix A.3.

## 5.2 MuJoCo Results

Table 1 presents a comparison of our TDIL method with the baselines in the single-demonstration setting. As expected, BC exhibits the least satisfactory results, which can largely be attributed to its dependence on extensive data for effective generalization. With limited training data, BC is prone to making incorrect decisions in unfamiliar states during testing, which leads to error accumulation (Ross et al., 2011). GAIL demonstrates comparable results to BC in all environments except *Hopper-v3*. Such an outcome may arise from the inherent instability associated with adversarial training, the sparse rewards produced within the Maximum Entropy IRL framework in the single-demonstration setting, and the algorithm's low sample efficiency. On the other hand, although designed under the assumption of environmental stationarity, f-IRL performs suboptimally across various tasks, and fails to function adequately on *Humanoid-v3*. This shortcoming is potentially attributable to the instability experienced during training. Our observations indicate that f-IRL's effectiveness diminishes as it approaches expert performance levels (please refer to our Appendix for more details). PWIL also fails to achieve expert-level performance, possibly due to its reliance on Euclidean distance, which does not adequately reflect the environmental dynamics. More specifically, when two state-action pairs are proximate in Euclidean distance but unreachable in the MDP, the computation of primal Wasserstein cost could yield inaccurate results. Although CFIL outperforms the expert's performance in *Ant-v3*, it does not sustain this advantage in *Walker2d-v3*, which highlights its limited generalization capabilities. In contrast, our TDIL consistently achieves expert-level performance across all tasks, demonstrating its superior adaptability and effectiveness.

## 5.3 Ablation Study

In this section, we analyze the impact of different components on the performance of the TDIL method. Specifically, we analyze TDIL with **(1)** different choices of the hyper-parameter $\beta$ used in Eq. 5 and **(2)** two variations: one that excludes the BC loss during training, and another that omits the use of hard negative samples $B_{\text{reversed}}^-$. The detailed results and analysis of **(1)** is shown in Table A5, it reveals that by setting $\beta$ within the range of [0.1, 0.9] the agent consistently achieves expert-level performance, without the use of Behavioral Cloning (BC) loss. This result highlights the effectiveness of the reward function $R_{\text{agg}}$.

Table 2: An ablation study of the TDIL method w/ and w/o the BC loss and hard negative samples.

| | Full TDIL Method | w/o BC | w/o Hard Neg. Samples | Expert Performance |
|---|---|---|---|---|
| HalfCheetah-v3 | $15,666 \pm 85$ | $12,630 \pm 6,854$ | $15,718 \pm 179$ | 15,251 |
| Hopper-v3 | $4,115 \pm 14$ | $3,890 \pm 562$ | $4,143 \pm 6$ | 4,114 |
| Ant-v3 | $6,434 \pm 66$ | $3,995 \pm 2,408$ | $6,571 \pm 116$ | 6,561 |
| Humanoid-v3 | $5,758 \pm 173$ | $5,575 \pm 196$ | $4,868 \pm 2,443$ | 5,855 |
| Walker2d-v3 | $6,312 \pm 47$ | $6,281 \pm 76$ | $6,268 \pm 53$ | 6,123 |

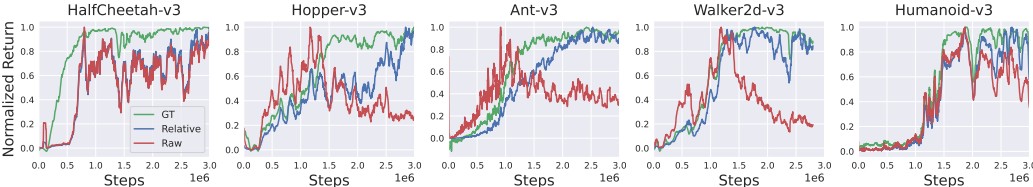

Figure 4: Comparison of normalized ground truth, raw, and relative rewards over 3M timesteps.

The results of **(2)** is reported in Table 2. The variant that omits the BC loss exhibits inferior performance relative to the fully-implemented TDIL method. This decline can be attributed to the absence of the BC loss, which provides essential guidance to the agent for achieving optimal policy in states that are proximal to the expert's states. On the other hand, the variant that omits the use of hard negative samples performs comparably well in all environments except for *Humanoid-v3*. This result suggests that, in simpler environments, the information contained in hard negative samples may not be necessary. However, in the *Humanoid-v3* environment, the absence of hard negative samples adversely affects performance. This discrepancy may be attributed to the vast state space of *Humanoid-v3*, which diminishes the chances that easy negative samples encapsulate the essential information contained in hard negative ones. Furthermore, as *Humanoid-v3* is a more delicate environment, the agent may be sensitive to inaccurately estimated rewards resulting from the absence of hard negative samples.

## 5.4 BLIND MODEL SELECTION

To validate relative return can be used in black model selection, we graph the ground truth return, raw return, as well as relative return obtained by the agent in an episode during training in Fig. 4. These rewards are all normalized by their respective maximum values over 3M timesteps. From these plots, a clear positive correlation between the relative rewards and the ground truth rewards can be observed across all environments, whereas the trends of the raw rewards obtained by the agents do not consistently align with the ground truth rewards. In particular, in *Hopper-v3*, *Ant-v3*, and *Walker2d-v3*, the raw rewards exhibit a trend that initially rises and then falls over 3M timesteps, a pattern that does not mirror the ground truth rewards. This fluctuation suggests that the accuracy of the transition discriminator grows during training, as it processes more expert data collected by the increasingly proficient RL agent. Additional results on blind model selection can be referred to Appendix A.4.5.

## 6 CONCLUSION

In this paper, we proposed TDIL as a robust methodology for addressing the challenges of single-demonstration IL. TDIL enriches the sparse reward landscape by incorporating a surrogate reward function, approximated by a transition discriminator, which guides the agent toward expert proximity states. According to our experimental results, TDIL exhibited remarkable optimality. When tested on the MuJoCo benchmarks, our method consistently delivered expert-level performance, outperforming all existing baseline algorithms, including BC, GAIL, f-IRL, PWIL, and CFIL. We also demonstrated the feasibility of TDIL in an Adroit Door task. Our work presents insights and a foundation for future advancements in single-demonstration IL, reinforcing its practical relevance in scenarios such as autonomous vehicles, and enhancing its stability, reliability, as well as practicality.

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

# A APPENDIX

In this appendix, we provide detailed training configurations, additional experimental results, and discussions on the proposed TDIL method. In Section A.3, we elaborate on the experimental setups as well as the model architecture adopted in our method. In Section A.4, we present additional experimental results to validate the effectiveness of our method.

## A.1 EXTENDED REVIEW OF RELATED WORK

The single-demonstration IL setting presents a unique and challenging problem domain. Earlier online IL research commonly treats this setting as a component of their ablation studies, and often overlooks its significance. This section engages in a discussion about several well-known online IL algorithms, which can be broadly grouped into two categories: adversarial-based and non-adversarial-based methods. This discussion allows us to highlight our differences from these prior methods, and delve into the primary reasons behind their limited effectiveness in the single-demonstration IL context.

**Adversarial-based methods:** Adversarial-based approaches aim to align the agent's state, state-action, or state-next-state distributions with those of the expert by employing various divergence or distance measures. For instance, GAIL (Ho & Ermon, 2016) adopts the GAN-like framework (Goodfellow et al., 2020) to train a discriminator and minimize Jensen-Shannon divergence. AIRL (Fu et al., 2018), on the other hand, utilizes forward KL-divergence to derive stationary rewards and enhance transfer learning. Building upon these approaches, the authors of Ke et al. (2021) and f-MAX (Ghasemipour et al., 2020) unify GAIL and AIRL under the umbrella of f-divergence. To further extend these methods, f-IRL (Ni et al., 2020) uses gradient descent to recover a stationary reward function from the expert density. In addition, recent research by the authors of Swamy et al. (2021) suggest that various forms of IL can be understood as moment matching under different assumptions. Another line of work is based on the DICE (Nachum et al., 2019) framework. For example, ValueDICE (Kostrikov et al., 2020) utilizes the Donsker-Varadhan formulation of KL-divergence to develop an off-policy method, while SparseDICE (Camacho et al., 2021) introduces a regularizer to enable training with sparse expert data. Inspired by ValueDICE, CFIL (Freund et al., 2023) trains a pair of normalizing flows to optimize the Donsker-Varadhan representation of KL-divergence. Moreover, a variety of research efforts (K. et al., 2019; Han et al., 2022; Zeng et al., 2022; Viano et al., 2022) have been directed towards addressing specific challenges within the field. For instance, DAC (K. et al., 2019) modifies GAIL to facilitate off-policy training and concurrently tackles reward bias issues. MD-AIRL (Han et al., 2022) enhances robustness by incorporating mirror-descent into AIRL. In a further effort to improve efficiency, both ML-IRL (Zeng et al., 2022) and $P^2$IL (Viano et al., 2022) have been designed to relax the nested policy evaluation and cost optimization loop. Most of the above methods, while being successful in online IL, do not perform well in the single-demonstration IL setting. The reason behind this can be attributed to two primary factors. The first factor is that the majority of their objectives typically align with a min-max formulation, which could lead to unstable training, especially in situations with limited data. The second factor is inherent to their distribution-matching nature, which necessitates taking expectations over the expert distribution. Nevertheless, this process could become unreliable when dealing with sparse expert data. In contrast to these previous approaches, our proposed methodology does not seek to match the distribution of the agent with that of the expert. This different approach avoids the issues of inaccurate expectations and unstable adversarial training.

**Non-adversarial based method:** Non-adversarial based methods often aim to circumvent unstable training by designating stationary rewards to guide the agent toward expert behavior. Examples include SQIL (Reddy et al., 2019), D2-Imitation (Sun et al., 2022), and ILR (Ciosek, 2022), which implement a binary reward scheme that assigns a value of $1$ to expert data and $0$ to agent data. These methods typically require a substantial amount of expert data to achieve optimal performance in practice. Another line of research explores a two-stage training approach, wherein a reward surrogate is first trained offline and then utilized during interaction with the environment. For instance, RED (Wang et al., 2019) estimates expert support by leveraging Random Network Distillation (Burda et al., 2019), while DRIL (Brantley et al., 2020) pretrains an ensemble of Behavior Cloning (BC) (Pomerleau, 1991) models and employs their variance as a cost function. EBIL (Liu et al., 2020) and NDI (Kim et al., 2020) employ density models, such as Energy-Based Models

(EBM) (Song & Kingma, 2021) and Masked Autoencoder Density Estimation (MADE) (Germain et al., 2015), to estimate expert support density. Nevertheless, these methods necessitate a significant amount of expert data for training the offline reward surrogate, which poses challenges when applied to the single-demonstration setting. Another non-adversarial approach, PWIL (Dadashi et al., 2021), attempts to minimize discrepancy between an agent's and an expert's distributions by employing the primal form of Wasserstein distance. This method requires the computation of the Euclidean distance between every state-action pair and those of the expert, a measure that may not precisely align with the distance as defined by the Markov Decision Process (MDP). In contrast, our method takes the properties of the underlying MDP into account. Furthermore, recent advancements such as IQ-Learn (Garg et al., 2021) and LS-IQ (Al-Hafez et al., 2023) offer a unique perspective, as they implicitly represent policy and reward using a single Q-function. Nevertheless, according to our experiments, these methods could suffer from instability during training and may not consistently perform well across various IL tasks.

## A.2 ALGORITHM AND TRAINING DETAILS

### A.2.1 PRACTICAL ALGORITHM

The training process concurrently updates the transition discriminator and the SAC agent. Both the agent and expert transition data are utilized to train the SAC agent, with the agent's reward calculated using the transition discriminator. The reward calculation method involves the computation of the reward of both agent data and expert data. The agent reward is calculated by pairing the next state $s_{t+1}$ of a transition $(s_t, a_t, s_{t+1})$ with 'every' expert state from the demonstration, as illustrated in Fig. 3, and using the transition discriminator to calculate the reachability probability of each pair. These probabilities are then summed to yield a reward $r_t = \sum_{i=0}^{T} D(s_{t+1}^a, s_i^e)$. The expert rewards are computed in a similar manner by pairing each next state of an expert transition with every other expert state, and summing the resulting probabilities.

### A.2.2 TRAINING STABILIZATION

To ensure stable training, a target transition discriminator, denoted as $\hat{D}$, is employed in our training process to compute the reward. $\hat{D}$ is soft-updated using the formula $\hat{D} = (1 - \lambda)D + \lambda\hat{D}$, where $\lambda$ is a hyperparameter set to 0.0001 in practice. The target transition discriminator helps mitigate the instability caused by SGD training, providing a more stable and consistent target for the SAC agent to learn from. This reduces overfitting and other potential sources of instability, making the training process less susceptible to fluctuations and ensuring a consistent trajectory towards convergence.

### A.2.3 ALGORITHM DETAIL

Algorithm 1 presents a practical training methodology of the proposed method, refering to TDIL. It takes as input the policy $\pi$ of an imitator agent, an environment $\mathcal{E}$, a replay buffer $B$, a Transition Discriminator $D$, a Target Transition Discriminator $\hat{D}$, and an expert trajectory $\tau^e$. The output is a trained optimal agent $\pi^*$. The training process is iterative, continuing until a convergence criterion is met. During each iteration, the policy $\pi$ interacts with $\mathcal{E}$, and the states and actions $(s_t, a_t, s_{t+1})$ are stored in $B$. Next, $D$ is updated based on Eq. (9) based on the stored transitions. Following this, $\hat{D}$ is soft-updated by $D$, which help stabilizing training. The algorithm then samples a batch of transitions from both $B$ and the expert trajectory $\tau^e$, and calculates the reward using $\hat{D}$. This reward is then used to update $\pi$ by comparing the agent's transitions with those of the expert. Finally, $\pi$ is updated using a BC loss, denoted as $L_{BC} = \text{MSE}(a \sim \pi(s_i^e), a_i^e)$, which aims to minimize the discrepancy between the agent's actions and the expert's actions. Through the repetition of these steps, the TDIL algorithm trains the imitator agent $\pi$ to match the expert's performance in the given environment. To satisfy the policy assumption in Section 2, the BC loss $L_{BC}$ is included to ensure $p(\mathcal{O}_t = 1|s_t) = \max_a p(\mathcal{O}_t = 1|s_t, a)$.

### A.2.4 COLOR CODING IN FIG. 1

The color-coding in Fig. 1(b)-(d) serves to visually represent the density of each reward signal. Meanwhile, in Fig. 1(f)-(h), the color-coding is intended to visualize the actions taken by the agent. This approach is analogous to techniques employed in optical flow research, where color is used

---

**Algorithm 1:** TDIL: IL via Transition Discriminator

---

**Input** : Imitator Agent $\pi$, Environment $\mathcal{E}$, Replay Buffer $B$, Transition Discriminator $D$,
Target Transition Discriminator $\hat{D}$, Expert Trajectory $\tau^e$
**Output:** Trained optimal agent $\pi^*$

1 **while** not converge **do**
2      $\pi$ interacts with $\mathcal{E}$, storing $s_t, a_t, s_{t+1}$ in $B$
3      Update $D$ with Eq. (9)
4      Soft-update $\hat{D}$ with $D$
5      Sample one batch of transition from $B$ and $\tau^e$, and calculate the reward with $\hat{D}$
6      Update $\pi$ using sampled agent transitions and expert transitions with calculated reward
7      Update $\pi$ with $L_{BC}$
8 **end**

---

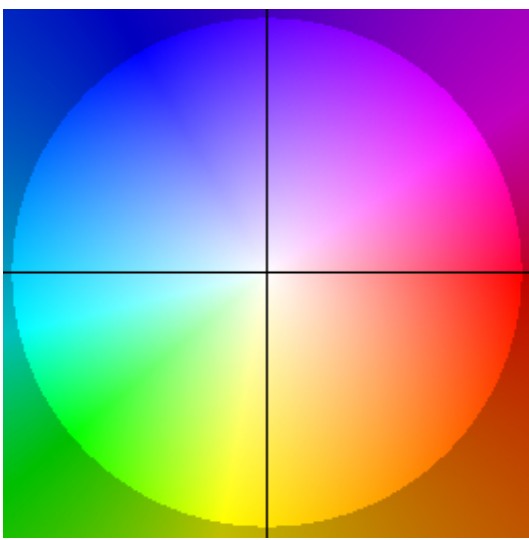

Figure A1: The relationship of color and direction used in the color-coding in Fig. 1 (f)-(h)

to provide an additional representation of direction beyond the traditional arrow visualization. To enhance clarity, the relationship between color and direction has been elucidated in Fig. Fig. A1, addressing the ambiguity and providing a more comprehensive understanding of the color-coded information in the figures.

### A.3 EXPERIMENTAL SETUPS

#### A.3.1 MODEL ARCHITECTURE OF TDIL

In this section, we provide the implementation details of TDIL. The backbone of TDIL is built upon the Soft Actor-Critic (SAC) framework. The actor and critic networks in SAC are implemented as neural networks with three hidden layers and rectified linear unit (ReLU) activation functions. Each of these hidden layers consists of 256 nodes. The actor's output is first projected to $[-1, 1]$ using a hyperbolic tangent (tanh) function, and then scaled to the value range required by the environments.

#### A.3.2 CODE IMPLEMENTATION AND HARDWARE CONFIGURATION

The code implementation and expert data used in this work are available on this anonymous repository. The computational requirements for the experiments presented in Section 5 is elaborated in Table A1.

Table A1: The hardware specification used to perform our experiments.

| Hardware | Specification |
|---|---|
| RAM | 128GB |
| CPU | AMD Ryzen Threadripper 3990X 64-Core Processor |
| GPU | NVIDIA GeForce RTX 3090 |

Table A2: Performance of baselines with BC loss.

|  | CFIL w/ BC | PWIL w/ BC | f-IRL w/ BC | TDIL w/ BC |
|---|---|---|---|---|
| HalfCheetah-v3 | 14853 | 4679 | 13638 | 15666 |
| Ant-v3 | 4683 | 5925 | 5337 | 6434 |
| Humanoid-v3 | 5343 | 5294 | N/A | 5758 |
| Walker2d-v3 | 6286 | 5489 | 4403 | 6312 |

## A.4 ADDITIONAL EXPERIMENTS

In this section, we provide additional experimental results and discussions. In Section A.4.1, we present the training curves of the proposed and the baseline methods to demonstrate the performance and stability of different algorithms. In Section A.4.5, we offer the evaluation results the models selected according to different blind selection metrics during training for demonstrating the effectiveness of the proposed blind selection method. Finally, in Section A.4.6, we examine the influences of the hard negative samples on the performance of the transition discriminators under various scenarios.

### A.4.1 TRAINING CURVES

Fig. A2 presents the training curves of TDIL as well as the other baseline methods, including BC, GAIL, f-IRL, PWIL, and CFIL. It is worth noting that the optimization of CFIL in the *HalfCheetah-v3* environment is numerically unstable as its output values sometimes become NaN during the training process. As a result, the training curve of CFIL in the *HalfCheetah-v3* environment can only be plotted partially. Fig. A2 demonstrates that TDIL is capable of reaching the expert level and exhibits a consistently stable training process across different environments compared to the other baselines.

### A.4.2 PERFORMANCE COMPARISON BETWEEN TDIL AND BASELINES WITH BC LOSS

We have conducted additional experiments to provide a more comprehensive analysis on adding BC loss into the training process of baselines. Table A2 presents the performance of CFIL, PWIL, and TDIL with BC loss, directly compared with training the agent with $R_{\text{TDIL}}$ and BC loss. Notably, some baselines demonstrate improved performance with BC loss, yet TDIL consistently outperforms all baselines. It is noteworthy that CFIL exhibited a substantial performance boost with BC loss in the Walker2d-v3 environment. However, it is crucial to acknowledge that CFIL encountered numerical issues, specifically the occurrence of actor output becoming NaN in the middle of training across all environments. This highlights potential instability in CFIL algorithm.

### A.4.3 EXPERIMENTS IN ADROIT HAND ENVIRONMENT

In Fig. A3, we present the experiment in the AdroitHandDoor environment. The AdroitHandDoor environment is a component of the Adroit manipulation platform, featuring a Shadow Dexterous Hand attached to a free arm with up to 30 actuated degrees of freedom. Introduced in (Rajeswaran et al., 2017) .

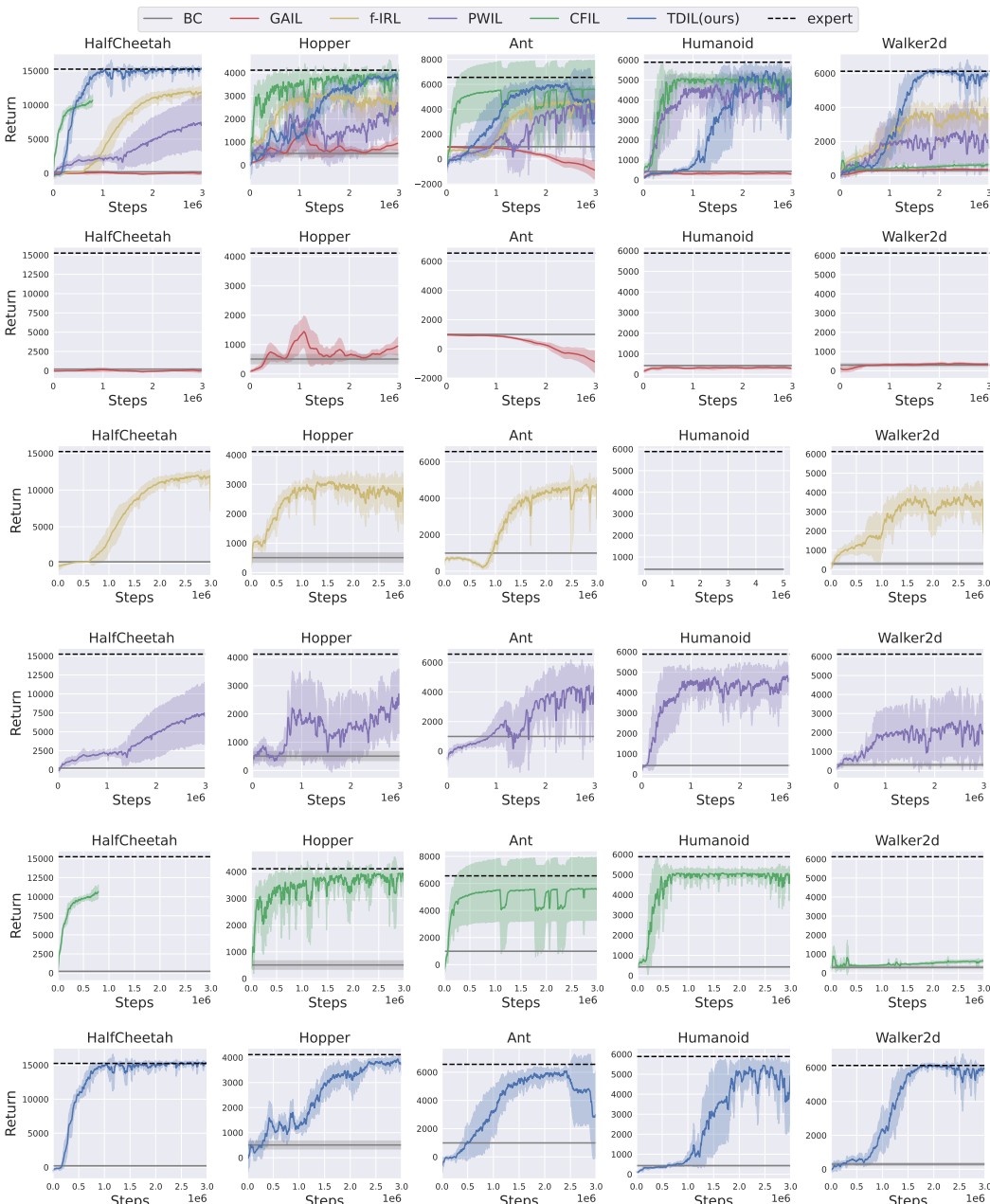

Figure A2: The training curves of BC, GAIL, f-IRL, PWIL, CFIL, and TDIL. These curves represent the means and the standard deviations of five independent runs conducted with different random seeds

In the AdroitHandDoor-v1 scenario, the task involves undoing a latch and swinging open a door with a biased torque that keeps it closed. The environment, based on a 28-degree-of-freedom system, includes a 24-degree-of-freedom ShadowHand and a 4-degree-of-freedom arm. The action space is represented as a Box(-1.0, 1.0, (28,), float32), with control actions specifying absolute angular positions of the hand joints. The observation space is a Box(-inf, inf, (39,), float64), containing information on finger joint angles, palm pose, and the state of the latch and door.

The episode's time step limit is set at 200. During the testing phase, the agent undergoes perturbation through five time-steps of random actions in the beginning of the episode to enhance difficulty and introduce stochasticity. In comparison to BC and two of the top-performing baselines from the main

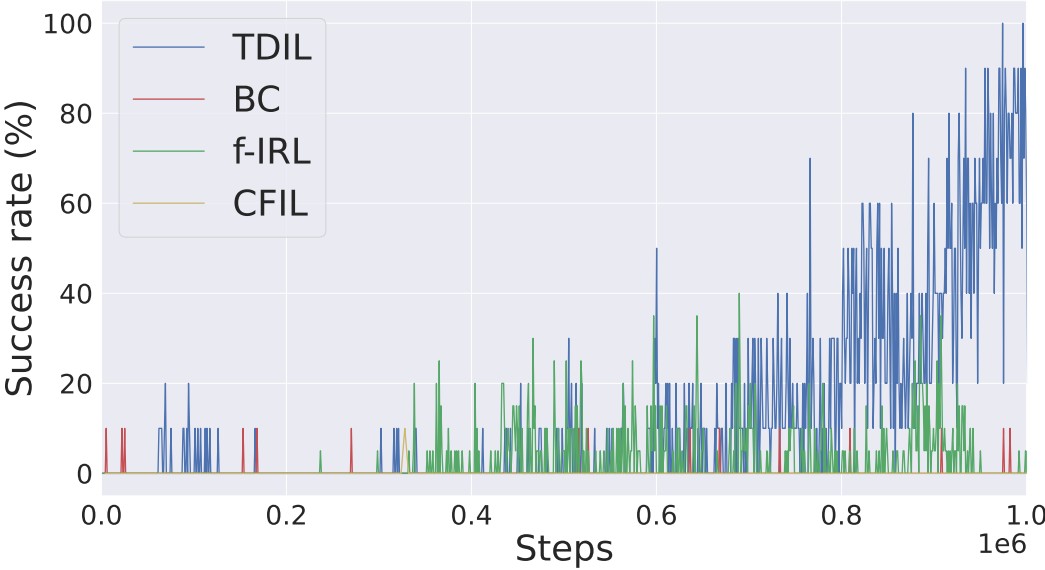

Figure A3: Comparing the success rates of TDIL, BC, f-IRL and CFIL in the AdroitHandDoor-v1 environment.

experiment, the results demonstrate that TDIL attains an expert-level performance within 1 million steps, surpassing the performance of BC, PWIL, and CFIL.

### A.4.4 EXPLORING RELATIVE REWARDS FOR BLIND MODEL SELECTION

Blind model selection refers to the process of choosing the optimal model checkpoint throughout the training phase, holds significant importance in the field of IL. In IL, it is generally assumed that obtaining the ground truth reward from the environment is unfeasible, even during testing. This issue, often neglected in prior research, warrants considerable attention. Although the reward signals proposed in this work, denoted as $R_{\text{TDIL}}$, can effectively train the agent, they may not be ideally suited for blind model selection. As training progresses, a potential decrease in the agent's raw rewards is observed. This decrease in raw agent reward does not necessarily indicate a decline in performance but may also reflect the improved accuracy of the transition discriminator. As a result, it becomes imperative to establish an indicator that is strongly correlated with the ground truth reward. Such an indicator would facilitate reliable model selection in IL. To meet this requirement, we introduce the concept of 'relative reward,' which is denoted as $r_{\text{relative}}$ and is defined as follows:

$$r_{\text{relative}} = r_{\text{raw agent}} / r_{\text{raw expert}}, \tag{A1}$$

where $r_{\text{raw agent}} = \sum_{t=0}^{\tilde{T}} R_{\text{TDIL}}(s_t, a_t)$ and $r_{\text{raw expert}} = \sum_{t=0}^{T} R_{\text{TDIL}}(s_t^e, a_t^e)$ are the total rewards along the agent's and expert's trajectories, and $\tilde{T}$ is the length of the agent's trajectory. As the transition discriminator may improve its accuracy during training, our aim is to mitigate the influence of its accuracy on reflecting the true extent of reward signals. In an ideal scenario, the reward for expert actions should be higher, while those outside the expert support should be lower. With this in mind, the essence of Eq. (A1) is to calculate the relative reward by dividing the raw agent reward, derived from the transition discriminator, by the raw expert reward, also derived from the transition discriminator. This process aids in neutralizing the impact of potential inaccuracies of the transition discriminator. The rationale behind this approach is the presumption that the inaccuracies in the transition discriminator would affect both the raw agent reward and the raw expert reward in a similar fashion. Hence, when the raw agent reward is divided by the raw expert reward, any inaccuracies that potentially exist in the transition discriminator should theoretically cancel out. This is because these inaccuracies are likely to proportionally affect the numerator (i.e., the raw agent reward) and the denominator (i.e., the raw expert reward) of the division. For example, if the transition discriminator is consistently underestimating or overestimating the rewards, both the raw agent reward and the raw expert reward would be underestimated or overestimated to a comparable extent. As a result,

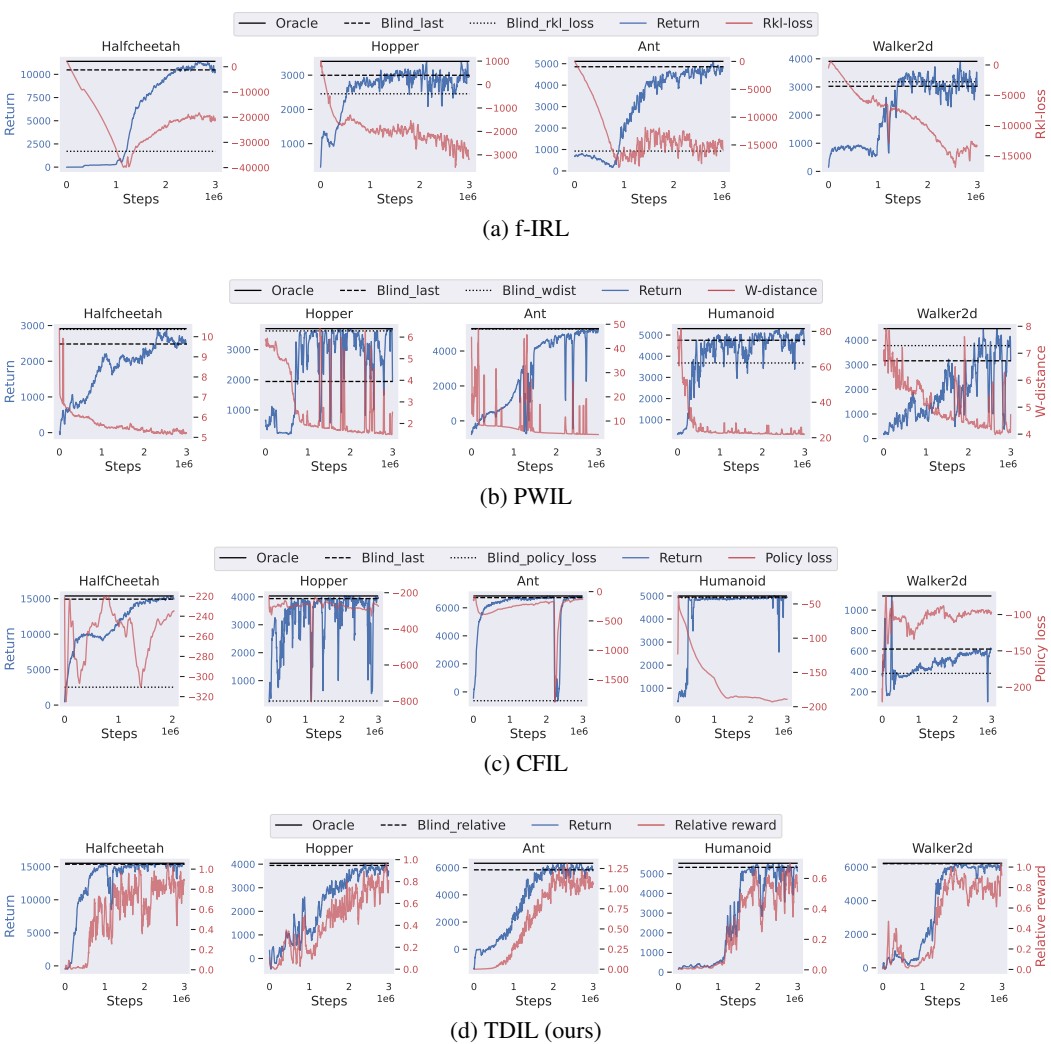

Figure A4: The training curves and the blind selection results of f-IRL, PWIL, CFIL, and TDIL (ours). The oracle line represents the highest evaluation return achieved during training. The Blind_last line depicts the evaluation return achieved by the agent at the end of the training phase. The Blind_{rkl_loss, wdist, policy_loss, relative} lines correspond to the evaluation returns determined based on the reverse KL loss, W-distance, policy-loss, and our proposed relative reward, respectively.

their ratio (i.e., the relative reward) should still provide a reliable comparison of agent performance relative to the expert, even if the absolute reward values are incorrect. This approach, therefore, helps to render the reward calculation more robust to the inaccuracies of the transition discriminator, and enhances the reliability of the model selection process in the single-demonstration IL context.

### A.4.5 BLIND MODEL SELECTION EXPERIMENTS

To further substantiate the efficacy of utilizing relative rewards in blind model selection, we performed a MuJoCo experiment in which the optimal testing model was selected without any access to the environmental ground truth rewards. In this experiment, our method used relative rewards as an indicator. In contrast, PWIL employed the Wasserstein distance, following the methodology of the original paper. For the remaining methods, which did not provide an indicator for model selection in their original manuscripts, we chose the model with the lowest policy loss. Table A3 presents the ratio of performance decrease of each method, which is calculated according

Table A3: Performance decrease ratios of different methods in the blind model selection scenario.

|  | BC (Pomerleau, 1991) | f-IRL (Ni et al., 2020) | PWIL (Dadashi et al., 2021) | CFIL (Freund et al., 2023) | Ours |
|---|---|---|---|---|---|
| HalfCheetah-v3 | $-0.75 \pm 0.34$ | $-0.27 \pm 0.27$ | $-0.17 \pm 0.15$ | $-0.07 \pm 0.00$ | **-0.02** $\pm 0.01$ |
| Hopper-v3 | $-0.27 \pm 0.23$ | $-0.32 \pm 0.26$ | $-0.31 \pm 0.29$ | **-0.04** $\pm 0.02$ | **-0.04** $\pm 0.05$ |
| Ant-v3 | $-0.73 \pm 0.03$ | $-0.18 \pm 0.16$ | $-0.13 \pm 0.18$ | **-0.03** $\pm 0.01$ | **-0.03** $\pm 0.01$ |
| Humanoid-v3 | $-0.18 \pm 0.16$ | N/A | $-0.18 \pm 0.16$ | **-0.03** $\pm 0.03$ | $-0.04 \pm 0.03$ |
| Walker2d-v3 | $-0.92 \pm 0.08$ | $-0.92 \pm 0.08$ | $-0.45 \pm 0.26$ | $-0.50 \pm 0.31$ | **-0.03** $\pm 0.04$ |

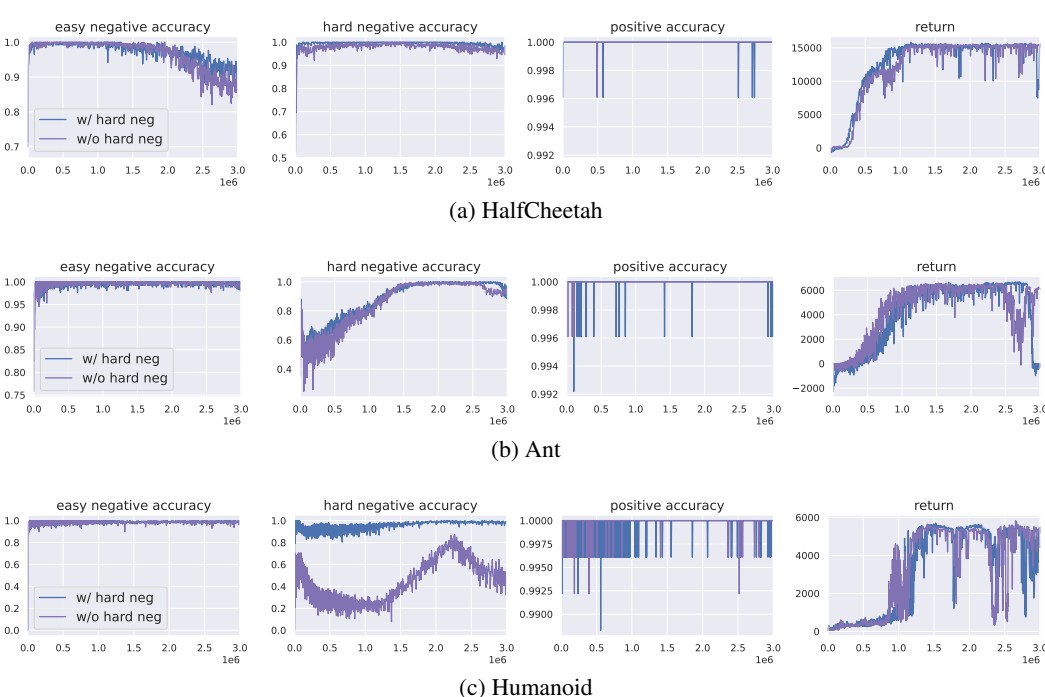

(a) HalfCheetah

(b) Ant

(c) Humanoid

Figure A5: An analysis of the accuracy of the transition discriminator under various scenarios.

to $\frac{blind\ result - oracle\ result}{oracle\ result}$. The results reveal that our proposed method outperforms both policy loss-based model selection and Wasserstein distance-based model selection schemes. This outcome suggests that relative rewards can effectively guide the selection of the best model, and provides a valuable insight that could be applied in future single-demonstration IL research to develop similar indicators for practical use.

To demonstrate the effectiveness of the blind model selection strategy over the model selection methods adopted by the baselines, we compare the returns obtained using the proposed strategy and the baseline methods along with the highest testing return achieved by each agent during its training process. Fig. A4 presents the results of the above setting. In the figure, the blue and red curves represent the total return obtained by each agent and the model selection strategy metric employed by each baseline, respectively. In addition, the solid and the dashed lines depict the highest testing return achieved by each agent during its training process and the return determined by the blind selection strategy, respectively. It is observed that our method is effective in selecting a model with high performance, as the distance between the solid and the dashed lines shown in Fig. A4 (d) is the closest as compared to those depicted in Figs. A4 (a), (b), and (c). Please note that the returns of the baseline methods can be derived using either the returns of the agent in the last step or the returns selected according to their respective blind selection metrics. In Table A3 of the main manuscript, we report the higher returns achieved by the baseline methods.

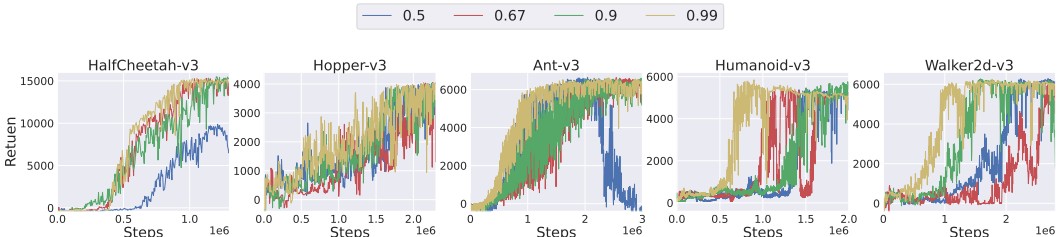

Figure A6: Performance of different $\alpha$ selection

Table A4: Accuracy of the transition discriminator when trained with different $\alpha$. "p" stands for positive data's accuracy; "en" stands for easy negative data's accuracy; "hn" stands for hard negative data's accuracy;

|  | 0.5(p) | 0.5(en) | 0.5(hn) | 0.67(p) | 0.67(en) | 0.67(hn) | 0.9(p) | 0.9(en) | 0.9(hn) | 0.99(p) | 0.99(en) | 0.99(hn) |
|---|---|---|---|---|---|---|---|---|---|---|---|---|
| HalfCheetah-v3 | 1.0 | 0.997 | 0.998 | 1.0 | 0.996 | 1.0 | 1.0 | 0.997 | 0.996 | 1.0 | 0.992 | 0.996 |
| Hopper-v3 | 0.996 | 0.996 | 0.992 | 0.996 | 0.996 | 0.996 | 0.996 | 0.996 | 0.992 | 1.0 | 0.996 | 0.996 |
| Ant-v3 | 0.996 | 1.0 | 1.0 | 1.0 | 1.0 | 1.0 | 1.0 | 0.996 | 1.0 | 1.0 | 0.992 | 0.992 |
| Humanoid-v3 | 1.0 | 1.0 | 1.0 | 1.0 | 0.996 | 0.996 | 0.996 | 0.996 | 0.996 | 1.0 | 0.99 | 0.988 |
| Walker2d-v3 | 0.996 | 0.996 | 0.996 | 0.992 | 0.996 | 0.996 | 0.996 | 0.992 | 0.99 | 1.0 | 0.992 | 0.988 |

### A.4.6 AN ANALYSIS OF THE ACCURACY OF THE TRANSITION DISCRIMINATOR

Fig. A5 illustrates the accuracy of the transition discriminator evaluated on positive samples, easy negative samples, and hard negative samples. Of particular interest is the accuracy of the hard negative samples. In *HalfCheetah-v3* and *Ant-v3* (i.e., Figs. A5(a) and (b), respectively), the transition discriminator trained without the use of hard negative samples demonstrates similar accuracy to the one trained with hard negative samples. However, in *Humanoid-v3* (Fig. A5(c)), the transition discriminator trained without hard negative samples exhibits significantly lower accuracy compared to the one trained with hard negative samples. These findings substantiate the assumption presented in Section 5.3, which suggests that the set of hard negative samples falls within the subset of easy negative samples. In relatively less complex environments, the agent can extract the information embodied in hard negative samples even when training exclusively with easy negative samples. However, this scenario is less probable in the more demanding *Humanoid-v3* environment, leading to the observed discrepancy in accuracy between the two training settings. These experimental results highlight the importance of incorporating hard negative samples, particularly in complex environments, to improve the accuracy and effectiveness of the proposed transition discriminator.

### A.4.7 SENSITIVITY ANALYSIS ON THE HYPER-PARAMETER $\alpha$

We have addressed the sensitivity of the proposed TDIL algorithm to different values of the hyper-parameter $\alpha$ by presenting the corresponding training curves in Fig. A6. The introduction of the balancing factor $\alpha$ for positive and negative samples aims to mitigate the impact of false negative samples within the pool of easy negative samples. These easy negative samples are composed of pairs of individually randomly sampled states from the replay buffer, and there exists a chance that these pairs may form valid transitions under the Markov Decision Process (MDP), effectively becoming positive samples.

To safeguard the training of the transition discriminator against the adverse effects of false negatives, we assign a smaller weight to negative samples compared to positive samples. Experimental results demonstrate that when $\alpha$ is set to small values (e.g., 0.5, 0.67), the agent takes longer to reach optimal performance in certain environments. Conversely, when the value of $\alpha$ is set to 0.99, the algorithm consistently performs well across various environments. This observation underscores the importance of choosing the hyper-parameter $\alpha$ to ensure optimal and robust performance of the TDIL algorithm.

Table A5: Performance of TDIL under different $\beta$ value selection.

| | 0+BC | 0 | 0.1 | 0.2 | 0.5 | 0.8 | 0.9 | 0.95 | 0.99 | 1.0 | Expert |
|---|---|---|---|---|---|---|---|---|---|---|---|
| HalfCheetah-v3 | **15666** | 12630 | 15100 | 15541 | 15479 | 15612 | 15289 | 15462 | 15529 | 9791 | 15251 |
| Hopper-v3 | 4115 | 3890 | 4124 | 4126 | **4162** | 4128 | 4083 | 1887 | 3232 | 1950 | 4114 |
| Ant-v3 | 6434 | 3995 | 6358 | 6513 | 6467 | 6611 | **6632** | 6560 | 6506 | 4216 | 6561 |
| Humanoid-v3 | 5758 | 5575 | 6288 | **6352** | 6312 | 6325 | 5680 | 5703 | 5235 | 1826 | 5855 |
| Walker2d-v3 | 6312 | 6281 | 6251 | 6204 | 6266 | 6346 | **6349** | 6296 | 6098 | 1769 | 6123 |

### A.4.8 EXPERIMENTAL RESULT ON DIFFERENT CHOICES OF HYPER-PARAMETER $\beta$

The ablation study on the hyper-parameter $\beta$ is comprehensively presented in Table A5, shedding light on its impact within the overall reward function. By aggregating $R_{\text{TDIL}}$ and $R_{\text{IRL}}$ with a judicious selection of $\beta$, the agent consistently attains expert-level performance guided by this composite reward. Experimental findings suggest that values of $\beta$ within the range of [0.1, 0.9] yield favorable results across various MuJoCo environments. This observation underscores the intrinsic ability and efficacy of the reward function $R_{\text{agg}}$.

Importantly, these results indicate that setting $\beta$ to zero, as done in the main experiments, can still produce effective outcomes, particularly when approximating the effect of $R_{\text{IRL}}$ through BC loss. This pragmatic approach not only maintains computational efficiency but also highlights the adaptability and robustness of the proposed TDIL method, even when certain components, such as $\beta$, are tuned or simplified for specific experimental contexts.

## A.5 MULTI-STEP EXPERT PROXIMITY

In the main manuscript, the expert reachability indicator $\tilde{\mathcal{O}}_t$ is only defined to consider the transition to expert states within a single timestep. We could generalize the reachability indicator to multiple timesteps by defining $\tilde{\mathcal{O}}_t^{(k)}$, where it determines whether the state $s_t$ can reach an expert state by selecting a series of $k$ actions. Formally, we define the following:

$$p(\tilde{\mathcal{O}}_t^{(k)} = 1 | s_t, a_t) \overset{\text{def}}{=} \begin{cases} \int_{\mathcal{S}} p(s_{t+1}|s_t, a_t) p(\mathcal{O}_{t+1} = 1|s_{t+1}) ds_{t+1} & \text{if } k = 1, \\ \int_{\mathcal{S}} p(s_{t+1}|s_t, a_t) p(\tilde{\mathcal{O}}_{t+1}^{(k-1)} = 1|s_{t+1}) ds_{t+1} & \text{if } k \in \{2, \ldots, T\}, \end{cases} \quad \text{(A2)}$$

The value of $p(\tilde{\mathcal{O}}_t^{(k)} = 1 | s_t)$ can be calculated as in the main manuscript. The surrogate reward functions corresponding to the indicators are defined as follows:

$$R_{\text{TDIL}}^{(k)}(s_t, a_t) \overset{\text{def}}{=} \mathbb{E}_{s_{t+1} \sim p(s_{t+1}|s_t, a_t)} \left[ p(\tilde{\mathcal{O}}_{t+1}^{(k)} = 1|s_{t+1}) \right]. \quad \text{(A3)}$$

Each surrogate reward functions can be approximated by $D^{(k)}(s_i, s_j)$ defined as:

$$D^{(k)}(s_i, s_j) \overset{\text{def}}{=} \max_{a_i, \ldots, a_{i+k-1}} \mathbb{1} \left[ \prod_{j=i}^{i+k-1} P(s_j|s_i, a_j) > 0 \right]. \quad \text{(A4)}$$

The total reward function $R_{\text{agg}}$ can then be re-defined as a weighted sum of the surroagte reward functions $R_{\text{TDIL}}^{(k)}$.

