# OpenReview forum: "Expert Proximity as Surrogate Rewards for Single Demonstration Imitation Learning"
_ICLR.cc/2024/Conference — Submitted to ICLR 2024_

### Official Review · Reviewer_6wvZ · 2023-10-24

**Soundness:** 3 good
**Presentation:** 3 good
**Contribution:** 3 good
**Rating:** 6
**Confidence:** 3

**Summary:**

The authors tackle the problem of single-demo imitation learning by introducing a reward function that rewards an agent for being at states within a single-step of an expert demonstration. They perform experiments on Mujoco navigation environments and 1 Adroit manipulation environment.

**Strengths:**

******************Writing:****************** Overall, the writing was solid and generally clearly explained the method and the equations used to derive the reward function.

****************************Motivation:**************************** The paper is addressing an important problem, that of limited-demonstration (or in their case, single) imitation learning.

******************Novelty:****************** To the best of my knowledge, the method is novel in this problem setting.

************************Experiments:************************ Results are moderately comprehensive and generally demonstrate that the authors’ method, TDIL, outperforms other methods by a decent amount.

**Weaknesses:**

**************Method:************** Some discussion is probably missing about how the method, combined with Q-learning, is likely to propagate values from the expert-proximal states to earlier states of the same trajectory.

************************Experiments:************************

- A missing ablation is one that ablates out $B^-_{\text{reversed}}$ as this seemed like an arbitrary choice when presented in Eq.9… i’d be curious to see what the performance without it looks like
- Since $\beta$ is set to 0 in the main experiments, there should be some ablation study on $\beta$ since it is proposed in the overall rew function of the method.
- The adroit experiments would also benefit from a comparison against at least another method other than BC for completeness

****************Clarity:****************

- Figure 1(a) is referenced in the intro yet doesn’t appear until page 4. To make the intro easier to read, it should be put earlier in the paper (it’s on page 4) or separated into multiple figures so that the relevant parts of Figure 1 appear when referenced in the intro.
- There’s some color-coding going on in Figure 1(b)-(d), (f)-(h) but they’re not explained in the figure or caption (presumably gray intensity in (b)-(d) is for the reward but I have no idea what the colors for (f)-(h) mean)
- Eq 6: In the 3rd line, I had no idea where $T$ came from (was it an MDP time horizon?) until I went back and saw in the preliminaries that the demonstration is assumed to be of length $T$. Would be helpful to specify this again here
- Generally, the use of lowercase $p(a|s$ to describe a policy is confusing because of the use of $p_0(s_0)$ to describe the initial transition dist. and uppercase $P$ for transition probabilities. I think it would be better to use $\pi(a|s)$ and $\pi^*(a|s)$ to describe policies and optimal policies, respectively.
- After Eq.6, it would be good to briefly clarify the use of actor-critic RL and TD-learning so that the expectation can safely be ignored.

**************************Minor Issues:**************************

- “that follows” → “that follow” between Eq3 and Eq4
- “To ensure optimality, the agent is trained with the total reward…” → “ensure” is probably not the right term here as only using $R_{IRL}$ is the only way to ensure optimality
- Fig2 should perhaps have an arrow pointing back from Step 4 to Step 1

**Questions:**

Why are the arrows in figure 1 curved? Are the direction arrows calculated based on the average direction represented by the logits for the discrete actions from the policy?

Does the assumption of optimality in the construction of $R_{TDIL}$ matter in practice? It makes the equations simpler, but in practice the reward function is more accurate if calculated under the expectation of the current policy.

Is Eq10 used for actual reward calculation for the policy or ********************only model selection********************? Is it even used at all for true model selection in the main experiments or do the authors simply pick the latest checkpoint for all methods?

---

> ### Author Response · Authors · 2023-11-22
> **Official Response for Reviewer 6wvZ**
>
> Dear reviewer 6wvZ, thank you for the positive feedback, detailed comments and constructive suggestions, we have addressed the concerns and suggestions as follows.
>
>
> ---
>
> **Comments**
> ---
>
> > **C1. :** Some discussion is probably missing about how the method, combined with Q-learning, is likely to propagate values from the expert-proximal states to earlier states of the same trajectory.
>
> **Response:** We appreciate the reviewer for bringing this to our attention. We have responded to this by incorporating relevant information at the end of Section 3.2 in our updated manuscript, in which we elaborated on how the proposed method, in conjunction with an agent (specifically the soft actor-critic (SAC) used in this paper), propagates values from states proximal to the expert to earlier states within the same trajectory.
>
> > **C2. :** A missing ablation study is one that ablates out $B^-_{Reversed}$ as this seemed like an arbitrary choice when presented in Eq.9… i’d be curious to see what the performance without it looks like.
>
> **Response:** We would like to bring to the reviewer’s kind attention that the ablation study presented in Table 2 of the original main manuscript, specifically in the column labeled “w/o Hard Negative Samples”. As discussed in Section 3.3, hard negative samples for the contrastive learning of the transition discriminator are denoted as $B^-{Reversed}$. These hard negative samples can consist of different combinations of data from reversed transitions, depending on prior knowledge about different environments. To maintain generality, we refer to these as hard negative samples throughout this paper, using the term to represent $B^-_{Reversed}$.
>
> The experiments that ablate hard negative samples demonstrate that in simpler environments, the information contained in these samples may not be necessary. In contrast, their absence in the Humanoid-v3 environment impairs performance. This difference can be attributed to the vast state space of Humanoid-v3, which reduces the likelihood that easy negative samples encapsulate the essential information contained in hard negative samples. Furthermore, given the complexity of Humanoid-v3, the agent may be sensitive to incorrectly estimated rewards that stem from the absence of hard negative samples.
>
> > **C3. :** Since $\beta$ is set to 0 in the main experiments, there should be some ablation study on it since it is proposed in the overall reward function of the method.
>
> **Response:** We appreciate the reviewer's suggestion and have augmented the ablation study on the hyperparameter $\beta$ in Table A3 of the supplementary material. The study reveals that by setting $\beta$ within the range of [0.1, 0.9] in $R_{agg} = \beta R_{IRL} + (1-\beta) R_{TDIL}$, the agent consistently achieves expert-level performance, **without the use of Behavioral Cloning (BC) loss**. This result validates efficacy of the proposed reward function $R_{agg}$.
>
> > **C4. :** The adroit experiments would also benefit from a comparison against at least another method other than BC for completeness.
>
> **Response:** We appreciate the reviewer's suggestion and have incorporated an experiment shown in Fig. A3 of the supplementary material. This includes comparisons with BC, CFIL, and f-IRL. The results demonstrate that TDIL achieves expert-level performance (i.e., a 100% success rate) within one million steps and surpasses the success rate of BC(10%), f-IRL(40%), and CFIL(10%).
>
> > **C5. :** Figure 1(a) is referenced in the intro yet doesn’t appear until page 4. To make the intro easier to read, it should be put earlier in the paper (it’s on page 4) or separated into multiple figures so that the relevant parts of Figure 1 appear when referenced in the intro.
>
> **Response:** We appreciate the kind suggestion from the reviewer. We have relocated Fig. 1 to page 2 to ensure it appears earlier in the paper for improving the alignment with references in the introduction.
>
> > **C6. :** There’s some color-coding going on in Figure 1(b)-(d), (f)-(h) but they’re not explained in the figure or caption (presumably gray intensity in (b)-(d) is for the reward but I have no idea what the colors for (f)-(h) mean)
>
> **Response:** We appreciate the reviewer's suggestion and have incorporated enhancements in the updated manuscript.
>
> The color-coding in Figs. 1 (b)-(d) serves to visually represent the density of each reward signal. Meanwhile, in Figs. 1 (f)-(h), the color-coding is intended to visualize the actions taken by the agent. This approach is analogous to techniques employed in optical flow research, where color is used to provide an additional representation of direction beyond the traditional arrow visualization.
>
> We also provide a reference in Section A.2.4 of the supplementary material to illustrate the relationship between color and direction. This is intended to offer a more comprehensive understanding of the color-coded information.

---

> > ### Author Response · Authors · 2023-11-22
> >
> > > **C7. :** Eq 6: In the 3rd line, I had no idea where $T$ came from (was it an MDP time horizon?) until I went back and saw in the preliminaries that the demonstration is assumed to be of length . Would be helpful to specify this again here
> >
> > **Response:** We appreciate the suggestion and have updated the manuscript to include a definition of $T$, which is typically referred to as the horizon and represents the total timesteps of the demonstration length.
> >
> > > **C8. :** Generally, the use of lowercase $p(a|s)$ to describe a policy is confusing because of the use of $p_0(s_0)$ to describe the initial transition dist. and uppercase $P$ for transition probabilities. I think it would be better to use $\pi(a|s)$ and $\pi^*(a|s)$ to describe policies and optimal policies, respectively.
> >
> > **Response:** We appreciate the constructive suggestion and have made the modification to the policy notation, changing it from $p(a|s)$ to $\pi(a|s)$.
> >
> > > **C9. :** After Eq.6, it would be good to briefly clarify the use of actor-critic RL and TD-learning so that the expectation can safely be ignored.
> > **C10. :** “that follows” → “that follow” between Eq3 and Eq4
> > **C11. :** “To ensure optimality, the agent is trained with the total reward…” → “ensure” is probably not the right term here as only using $R_{\text{IRL}}$ is the only way to ensure optimality.
> > **C12. :** Fig2 should perhaps have an arrow pointing back from Step 4 to Step 1
> >
> > **Response:** We are grateful for the constructive suggestions and have accordingly revised the manuscript to reflect these valuable inputs. To address C11, the phrase has been revised to "To ensure optimality on the expert's states……" to better reflect the intended meaning.
> >
> >
> > **Questions**
> > ---
> >
> > > **Q1. :** Why are the arrows in figure 1 curved? Are the direction arrows calculated based on the average direction represented by the logits for the discrete actions from the policy?
> >
> > **Response:** We appreciate the reviewer's question regarding Fig. 1. The curved arrows in the figure represent the average direction calculated from the logits for the discrete actions produced by the learned policy at various grid locations. To enhance clarity, we have included this explanation in the caption of Fig. 1 in the updated manuscript.
> >
> > > **Q2. :** Does the assumption of optimality in the construction of matter in practice? It makes the equations simpler, but in practice the reward function is more accurate if calculated under the expectation of the current policy.
> >
> > **Response:** Thanks for raising this question.  In practice, it is also important to ensure the agent is optimal on the expert states. As evidenced in Table A4 of the supplementary material, agents with optimality ensured by BC loss (column $\beta = 0$ + BC) or IRL reward consistently (columns $0 < \beta < 1$)  outperform those that only use $R_{TDIL}$ as the reward (column $\beta=0$). This emphasizes the practical significance of the optimality assumption in achieving expert performance.
> >
> > > **Q3. :** Is Eq10 used for actual reward calculation for the policy or only model selection? Is it even used at all for true model selection in the main experiments or do the authors simply pick the latest checkpoint for all methods?
> >
> > **Response:** We would like to bring to the reviewer’s kind attention that Eq. (10) is only employed for blind model selection experiments. For all other experiments, including the main experiments, we use the ground truth reward to select the best checkpoint for all the algorithms.
> >
> > A detailed presentation of the results and analysis from the blind model selection experiments is available in Sections A.4.4 and A.4.5. It is important to note that the model selected blindly based on our method consistently exhibits the lowest performance decrease ratio compared to the baseline algorithms.

---

### Official Review · Reviewer_eL6k · 2023-10-28

**Soundness:** 2 fair
**Presentation:** 3 good
**Contribution:** 2 fair
**Rating:** 5
**Confidence:** 4

**Summary:**

The paper proposes an expert proximity reward to improve the performance of existing Inverse RL and Adversarial IL algorithms. The intuition is based on a heuristic idea that the policy should go towards the expert transition in each step. The proposed algorithm is compared to various recent IRL/AIL algorithms on the Mujoco benchmark and can achieve better performance.

**Strengths:**

1. The proposed idea is simple.
2. The derivation of each step is in general clear.

**Weaknesses:**

1. The novelty of the idea. In may existing works, people have discussed method that can try to encourage the agent to go back to the expert distribution such as FIST [1]. Among IRL methods, PWIL and OT[2] also incorporates a dense distance-based reward that encourage the agent to stay close to the expert per-transition. I am wondering why the proposed method is better than these prior approaches.

2. Soundness of the method. If the agent has gone off the expert distribution, the proposed reward function can still degenerate. For example, if the minimal distance from a state $s_t$ to any expert state is 5 steps, the proposed reward $r(s_t, a_t)$ will output 0 no matter what action $a_t$ the policy chooses. Then how could the proposed method guide the policy?

3. Unfair comparison. The proposed method integrates BC in its training. However, it looks like all the baselines the authors compare to does not involve BC in training. This makes the comparison unfair.

4. Sensitivity analysis. The proposed algorithm involve a hyper parameter $\alpha$, but the authors do not discuss the sensitivity of the performance with respect to different alphas.

[1] Hakhamaneshi et al. HIERARCHICAL FEW-SHOT IMITATION WITH SKILL TRANSITION MODELS. NeurIPS 2021.
[2] Haldar et al. Supercharging Imitation with Regularized Optimal Transport. CoRL 2022.

**Questions:**

See weaknesses part.

---

> ### Author Response · Authors · 2023-11-22
> **Official Response for Reviewer eL6k**
>
> Dear reviewer eL6K, thank you for the constructive comments and feedback, we have addressed your concerns as follows.
>
>
> ---
>
> **Comments**
> ---
>
> >**C1. :** In many existing works, people have discussed methods that can try to encourage the agent to go back to the expert distribution such as FIST [1]. Among IRL methods, PWIL and OT [2] also incorporate a dense distance-based reward that encourages the agent to stay close to the expert per-transition. I am wondering why the proposed method is better than these prior approaches.
>
> **Response:** We would like to thank the reviewer for raising these questions.
>
> Although prior approaches such as FIST [r1], PWIL [r2], and OT [r3] adopt strategies to motivate agents to return to the expert distribution, the distance measuring functions they employ, including Euclidean and Cosine distances, are not theoretically guaranteed to achieve this objective. Furthermore, if the reward function does not consider the environment dynamics, it may mistakenly direct the agent to states that are not genuinely “close”, and may lead to the agent becoming stuck in those states. Therefore, the awareness of environmental dynamics is crucial for the reward function. Consider the toy example in Fig. 1 (a) (discussed in Section 3). When using the L2 distance (i.e., Euclidean distance) as the metric for measuring closeness, the reward system would inadvertently encourage the agent towards all adjacent states of the expert states, even if a barrier (indicated by a red line) separates the agent’s current state from the expert state. As a result, as shown in Fig. 1 (f), even after training, the cells on the inner side of the L-shaped barrier still exhibit an incorrect policy (pointing to the bottom left).
>
> To address these issues, a key insight of our paper is the use of a transition discriminator to obtain an environment-dynamic-aware reachability metric. This metric does not rely on the L2 distance. Instead, our transition discriminator considers transition reachability and thus is capable of providing more meaningful guidance for the agent to transition to the expert support and achieve expert-level performance in various environments.
>
> In addition, our breakthrough lies not just in introducing a more theoretically sound distance measuring function but also in densifying the single demonstration from a one-dimensional manifold to an *n*-dimensional support. This enhancement facilitates the alignment of the demonstration with the agent support. The significance of this lies in the fact that a demonstration restricted to a 1-manifold does not overlap with the agent support in a low-dimensional manifold, as discussed in [r4]. We have elaborated on this in the updated version of Section 3.1 in our manuscript.
>
> [r1] Hakhamaneshi et al. Hierarchical Few-Shot Imitation with Skill Transition Models. NeurIPS 2021.
>
> [r2] Dadashi, Robert, et al. Primal Wasserstein imitation learning. arXiv:2006.04678 (2020).
>
> [r3] Haldar et al. Supercharging Imitation with Regularized Optimal Transport. CoRL 2022.
>
> [r4] Arjovsky, Martin, and Léon Bottou. Towards principled methods for training generative adversarial networks. arXiv:1701.04862 (2017).
>
> >**C2. :** If the agent has gone off the expert distribution, the proposed reward function can still degenerate. For example, if the minimal distance from a state to any expert state is 5 steps, the proposed reward will output 0 no matter what action the policy chooses. Then, how could the proposed method guide the policy?
>
> **Response:** We would like to thank the reviewer for raising this question.
>
> We wish to draw the reviewer’s attention to the fact that for states multiple steps away from expert proximity, RL methods like the soft actor-critic (SAC) used in this work can propagate the rewards to other states. This propagation occurs through value functions and the agent's interaction with the environment via exploration and exploitation.
>
> A key insight of our method is the densification of the single demonstration’s 1-manifold into a theoretically sound *n*-dimensional support. This approach enables the RL agent to reach those states during exploration.
>
> On the other hand, while methods like PWIL provide rewards for more states, they lack theoretical soundness. This is due to their use of Euclidean (L2) or Cosine distance measurements, which do not account for environmental dynamics, as discussed in Section 3.1. In addition, while our method has already achieved optimal performance in our experiments with a 1-step setup, it retains the flexibility to extend to $n$ steps if necessary. This extension can be achieved by augmenting $(s_t, s_{t+m}), 1 \leq m \leq n$, into the positive example set $B^+$. The theoretical details are provided in Section A5 of our supplementary material.

---

> ### Author Response · Authors · 2023-11-22
>
> >**C3. :** The proposed method integrates BC in its training. However, it looks like all the baselines the authors compare do not involve BC in training. This makes the comparison unfair.
>
> **Response:** We would like to clarify the question as follows.
>
> Firstly, consider the scenario where BC loss is utilized. Table r1 presents the performance of CFIL, PWIL, f-IRL, and TDIL with BC loss. It is observed that although some baselines have improved their performance with BC loss (while others worsened), TDIL consistently outperforms all these baselines.
>
> Second, we wish to highlight that the proposed reward function is actually $R_{agg} = \beta R_{IRL} + (1 - \beta)R_{TDIL}$ (refer to Eq. (5) in the manuscript). Using $R_{agg}$, our **proposed method can train an SAC agent without employing BC**. The results, as reported in Table r2 (which corresponds to Table A4 in the supplementary material), reveal that even without BC, our agent can easily reach expert performance for $\beta \in [0.1, 0.9]$.
>
> Considering these two perspectives, it becomes evident that the proposed surrogate reward $R_{TDIL}$ can effectively assist the agent in consistently reaching expert performance. This is achievable as long as the agent is trained to perform the expert action on the demonstration since $R_{TDIL}$ only motivates the agent to move toward the expert’s proximity support and does not dictate specific actions to be performed on the expert states.
>
> - **Table r1** Performance of baselines with BC loss
>
> |   | HalfCheetah-v3 | Walker2d-v3 | Humanoid-v3 | Ant-v3 |
> | - | - | - | - | - |
> |**Expert** | 15251 | 6123 | 5855 | 6561 |
> | **TDIL with $R_{agg}$ <br> (w/o BC)** | 15612 | 6349 | 6352 | 6632|
> | **TDIL with BC** | **15666** | **6312** | **5758** | **6434**|
> | **CFIL with BC** | 14853 | 6286 | 5343 | 4683 |
> | **PWIL with BC** | 4679 | 5489 | 5294 | 5925|
> | **f-IRL with BC** | 13638 | 4403 | -      | 5337 |
>
>
>
> - **Table r2** Performance of TDIL under different $\beta$ value selection.
>
> |            	| 0+BC  | 0	| 0.1   | 0.2   | 0.5   | 0.8   | 0.9   | 0.95  | 0.99  | 1.0  | Expert |
> | -------------- | ----- | ---- | ----- | ----- | ----- | ----- | ----- | ----- | ----- | ---- | ------ |
> | HalfCheetah-v3 | **15666** | 12630 | 15100 | 15541 | 15479 | 15612 | 15289 | 15462 | 15529 | 9791 | 15251 |
> | Hopper-v3  	| 4115  | 3890 | 4124  | 4126  | **4162** | 4128  | 4083  | 1887  | 3232  | 1950 | 4114   |
> | Ant-v3     	| 6434  | 3995 | 6358  | 6513  | 6467  | 6611  | **6632** | 6560  | 6506  | 4216 | 6561   |
> | Humanoid-v3	| 5758  | 5575 | 6288  | **6352** | 6312  | 6325  | 5680  | 5703  | 5235  | 1826 | 5855   |
> | Walker2d-v3	| 6312  | 6281 | 6251  | 6204  | 6266  | 6346  | **6349** | 6296  | 6098  | 1769 | 6123   |
>
>
> >**C4. :** Sensitivity analysis. The proposed algorithm involves a hyperparameter but the authors do not discuss the sensitivity of the performance with respect to different alphas.
>
> **Response:** We would like to thank the reviewer for the suggestion concerning the sensitivity of performance to the hyperparameter $\alpha$. To address this, we have included two additional results in the revised manuscript.
>
> - Table A4 in the supplementary material presents the accuracy of transition discriminators trained with different values of $\alpha$. The results demonstrate that all setups of the transition discriminator can achieve an accuracy of over 0.988 for both positive and negative examples.
> - Furthermore, Fig. A6 in the supplementary material illustrates the training curve of the agent with different choices of $\alpha$. These results indicate that the majority of the $\alpha$ settings can attain expert performance.
>
> Drawing on the findings from these experiments, we can infer that the hyperparameter $\alpha$ is relatively straightforward to calibrate, indicating its robustness and ease of adjustment in different scenarios.

---

### Official Review · Reviewer_9w53 · 2023-10-31

**Soundness:** 2 fair
**Presentation:** 2 fair
**Contribution:** 2 fair
**Rating:** 5
**Confidence:** 3

**Summary:**

This work focuses on the single-demonstration imitation learning setting. This work highlighted a challenge of single-demonstration imitation, i.e., scarce expert data may result in sparse reward signals. To solve this issue, this work proposes to augment reward signals using a transition discriminator. The resulting algorithm is called transition discriminator-based imitation learning (TDIL).

**Strengths:**

1.	This work aims to use a transition discriminator to approximate surrogate rewards to enhance the convergence speed in the single-demonstration IL setting. The transition discriminator is introduced to capture the awareness of the environmental dynamics. The effectiveness of TDIL is evaluated in five MuJoCo tasks.

**Weaknesses:**

1.	The presentation of the paper is not easy to follow, and sometimes confusing. For example, what is the difference between the single demonstration and sparse/scarce expert data, which one is the source of the sparse reward issue in single-demonstration imitation learning?

2.	This work highlighted that the scarce expert data in single-demonstration imitation learning may result in sparse reward signals. However, the motivation example is not convincing. Section 3.1 claimed that the issue of sparse rewards is due to the difference in convergence rates originates from IRL’s particular feature of allocating rewards solely to states that mirror those of the experts. There is no evidence to show the relationship between the convergence speed and sparseness of the rewards. Moreover, there are no statistical results to directly show the sparseness of the IRL’s rewards.

3.	TDIL aims to use the expert reachability indicator to define a denser surrogate reward function. However, the indicator can only output binary values for state-action pairs, which will result in sparse rewards. It is confusing how such an indicator can be used to define a denser surrogate reward function. A potential explanation in section 3.2 is that the surrogate reward function encourages the agent to navigate to states that are in expert proximity. What is expert proximity?

4.	The transition discriminator is trained in a way that is like contrastive learning. It can capture the transition dynamics because the input data is state transition pairs.  How the awareness of the environmental dynamics can help it output denser rewards is unclear.

**Questions:**

1.	See weakness.

---

> ### Author Response · Authors · 2023-11-22
> **Official Response for Reviewer 9w53**
>
> Dear 9w53, thank you for the constructive comments and feedback, we have addressed your concerns as follows.
>
> ---
>
> **Comments**
> ---
>
> >**C1. :** What is the difference between the single demonstration and sparse/scarce expert data, which one is the source of the sparse reward issue in single-demonstration imitation learning?
>
> **Response:** As discussed in Section 1 and mentioned in the other sections of the main manuscript, single demonstration imitation learning (abbreviated as single demo IL) differs from conventional IL. In conventional IL, an agent typically has access to a number of expert demonstrations, which could range from hundreds to thousands, as in the case of D4RL [r1]. In contrast, the most challenging aspect of single demo IL is the limitation to only **one** trajectory accessible to the learning agent as a demonstration. This challenge arises because obtaining expert demonstrations is often expensive and can incur prohibitive efforts. In such a single demo setting, the demonstration is sparse compared to conventional IL settings. Utilizing conventional inverse reinforcement learning (IRL) approaches in the single demo setting would result in a sparse reward scenario.
>
> >**C2. :** There is no evidence to show the relationship between the convergence speed and sparseness of the rewards. Moreover, there are no statistical results to directly show the sparseness of the IRL’s rewards.
>
> **Response:** We would like to address the reviewer's question as follows. Consider a continuous environment with *n* dimensions. From a mathematical perspective, the reward sparseness of conventional IRL compared to TDIL can be described as follows:
> - For conventional IRL, one demonstration trajectory corresponds to only a 1-manifold (i.e., a 1-dimensional line).
> - In contrast, the TDIL method includes all states that are one-step reachable from the single demonstration trajectory. As a result, the reward function formulated by TDIL corresponds to an *n*-dimensional support and is infinitely denser than any 1-manifold [r2].
> A crucial aspect is to ensure that the demonstration is a support, as it is impossible to match an agent's support with a 1-manifold. We have enhanced Section 3.1 in the updated manuscript to improve accessibility and readability.
>
> >**C3. :** The indicator can only output binary values for state-action pairs, which will result in sparse rewards. It needs to be clarified how such an indicator can be used to define a denser surrogate reward function. A potential explanation in section 3.2 is that the surrogate reward function encourages the agent to navigate to states that are in expert proximity. What is expert proximity?
>
> **Response:** We would like to address the reviewer's question as follows.
>
> As discussed in Section 1 and defined in Section 3, **expert proximity** is a set of states where each state in this set can reach an expert’s demonstration state in one step. As discussed in Section 3, TDIL provides reward to the state-action pair $(s_t, a_t)$ if $s_{t+1}$ is in expert proximity.
>
> Please note that outputting binary values for state-action pairs does not affect the reward density. The transition discriminator only tells the one-step reachability. The sparseness or denseness of the reward system only depends on the size of the manifold that can offer non-zero rewards.  We increase the density by densifying the non-zero reward from a one-dimensional manifold to an *n*-dimensional manifold. For example, in a high-dimensional continuous environment such as Mujoco, there will only be at most 1,000 state-action pairs provided by expert demonstrations. However, the states in expert proximity are infinite. As a result, the number of state-action pairs that can reach a state in expert proximity is also infinite. We have enhanced Section 3.2 in the updated manuscript for improved accessibility.
>
> [r1] Fu, Justin, et al. "D4rl: Datasets for deep data-driven reinforcement learning." arXiv preprint arXiv:2004.07219 (2020).
>
> [r2] Arjovsky, Martin, and Léon Bottou. "Towards principled methods for training generative adversarial networks." arXiv preprint arXiv:1701.04862 (2017).

---

> > ### Author Response · Authors · 2023-11-22
> >
> > >**C4. :** The transition discriminator is trained in a way that is like contrastive learning. It can capture the transition dynamics because the input data is state transition pairs. How the awareness of the environmental dynamics can help it output denser rewards is unclear.
> >
> > **Response:** The purpose of our densified surrogate reward is to guide the agent back to an expert support. This proximity enables the agent to actually reach the expert demonstration in accordance with the environmental dynamics, and allows the agent to subsequently follow the demonstration. This aspect is essential because if the derived reward function does not consider the environment dynamics, it may mistakenly direct the agent to states that are not genuinely “close”, which may lead to the agent becoming stuck in those states. As a result, being aware of the environmental dynamics is crucial to **ensure the soundness** of our densified surrogate reward (not the density).
> >
> > Consider the toy example in Fig. 1 (a) (discussed in Section 3). Employing the L2 distance (i.e., the Euclidean distance) as the metric for measuring closeness, the reward system would encourage the agent towards all adjacent states of the expert states, even when a barrier (indicated by a red line) exists between the agent’s current state and the expert state. As a result, as depicted in Fig. 1 (f), even after training, the cells on the inner side of the L-shaped barrier still exhibit an incorrect policy (pointing to the bottom left). On the other hand, our method provides correct policy (Fig. 1 (h)) on all the cells. Showing the soundness of our reward function.

---

### Official Review · Reviewer_Ka6u · 2023-11-03

**Soundness:** 2 fair
**Presentation:** 3 good
**Contribution:** 3 good
**Rating:** 5
**Confidence:** 4

**Summary:**

The authors propose a method for single demonstration imitation learning which learns a discriminator that generates a pseudo reward by rewarding an agent for producing a state in the environment from which entering any state in an expert demonstration is possible. The pseudo reward is used to encourage the learned policy to be able to enter one of the states in the demonstration. The policy is also trained on a BC loss encouraging it to remain within the state distribution of the expert demonstration. The authors test their method on 5 mujoco environments.

**Strengths:**

I had the pleasure of reviewing this paper as a submission to an earlier venue and I must congratulate the authors on this improved version.
1. The method is principled and intuitive
2. The paper is well-written and organized.
3. The problem being addressed is crucial and the analysis includes model selection which is especially useful for deployment in real-world scenarios.

**Weaknesses:**

I still think a couple of points need to be improved upon before this is ready to be published:
1. **Additional experiments on a larger domain set**: Locomotion environments tend to be somewhat easier for RL agents (evidenced by a variety of self-supervised methods learning to walk). I would recommend increasing the scope of the experiments to a robotic manipulation domain which would allow readers to understand the limitations of the work.
2. **Ablations utilizing high dimensional states**: How does the method perform when utilizing demonstrations from the visual domain? This is necessary for completeness as there are several recent works for instance FISH https://arxiv.org/abs/2303.01497, RoboCLIP https://arxiv.org/pdf/2310.07899.pdf which learn a reward from visual observations from a single demonstration.

**Questions:**

I would be happy to increase my score if the authors can resolve the above questions.

---

> ### Author Response · Authors · 2023-11-22
> **Official Response for Reviewer Ka6u**
>
> Dear Reviewer Ka6u, thank you for your positive feedback and encouraging review. We have addressed your concerns as follows:
>
>
> ---
>
> **Comments**
> ---
>
>
> > **C1. Additional experiments on a larger domain set:** Locomotion environments tend to be somewhat easier for RL agents (evidenced by a variety of self-supervised methods learning to walk). I would recommend increasing the scope of the experiments to a robotic manipulation domain which would allow readers to understand the limitations of the work.
>
> **Response:** We would like to thank the reviewer for raising this question. We wish to draw the reviewer’s attention to Section A.4.3 and Fig. A3 (or Section 5.4 and Fig. A4 in the old version), where we presented the experiment in the AdroitHandDoor environment [r1]. This environment is a task of the Adroit manipulation platform, which features a Shadow Dexterous Hand attached to a free arm with up to 30 actuated degrees of freedom. The results demonstrated that the proposed TDIL method attained expert-level performance (100% success rate) within one million steps and was able to surpass the performance of BC(10% success rate), CFIL(10% success rate), and f-IRL(40% success rate).
>
> >**C2. Ablations utilizing high dimensional states:** How does the method perform when utilizing demonstrations from the visual domain? This is necessary for completeness as there are several recent works, for instance FISH https://arxiv.org/abs/2303.01497, RoboCLIP https://arxiv.org/pdf/2310.07899.pdf which learn a reward from visual observations from a single demonstration.
>
> **Response:**
> Thank you for mentioning these two highly related papers. We have cited them in our latest version.
> We would like to address this question by elaborating on the differences in settings between these methods and ours.
>
> In the case of RoboCLIP [r2], the reward signal was generated using a **pretrained** video-and-language model (VLM) that was trained with additional data. In contrast, our proposed TDIL method trains the agent from **scratch**.
>
> Regarding FISH [r3], it employed cost functions such as Euclidean (L2) and Cosine distance to derive rewards. However, such cost functions **are not theoretically guaranteed** since they do not consider the dynamics of the environment. For example, two states might be close in Eurclidean (L2) distance, but they can not reach to each other quickly since there are barriers between them.
> In contrast, our reward function is specifically designed to capture these dynamics, which is a primary focus of our paper. By considering dynamics, TDIL achieves superior performance, as demonstrated in our motivational experiment (please kindly refer to Fig. 1) and in the experiments detailed in Section 5 (Note that PWIL uses L2).
>
> In this paper, our main focus is to introduce a transition discriminator-based approach to enhance reward density and facilitate one-demonstration imitation learning scenarios. In the future, exploring the leverage of high-dimensional inputs (e.g., images) would be an interesting direction to investigate.
>
> [r1] Rajeswaran, Aravind, et al. Learning Complex Dexterous Manipulation with Deep Reinforcement Learning and Demonstrations. arXiv:1709.10087 (2017).\
> [r2] Sontakke, Sumedh A., et al. Roboclip: One Demonstration is Enough to Learn Robot Policies. arXiv:2310.07899 (2023).\
> [r3] Haldar, Siddhant, et al. Teach a Robot to FISH: Versatile Imitation from One Minute of Demonstrations. arXiv:2303.01497 (2023).

---

### Author Response · Authors · 2023-11-23
**Summary of our responses**

Dear area chairs and reviewers,

We appreciate the constructive feedback from all the reviewers and would like to summarize our primary responses as follows.

- Compared to conventional IRL methods such as GAIL, TDIL enhances the single demonstration IL by expanding the demonstration from a one-dimensional manifold to a multi-dimensional support. This enhancement is both essential and crucial, as a one-dimensional manifold does not overlap with the agent support and can lead to a zero training signal when the discriminator is perfect, as noted in [r1].

- Although there exist other dense reward IRL methods (such as PWIL and FISH) that are designed to guide the agent to states that are “close” to the expert states, the distance metrics they employed, such as the Euclidean (L2) or the Cosine distance metrics, are not theoretically sound. For example, using the L2 distance, two adjacent grid cells would be considered 'close' to each other even if a barrier exists between them. On the other hand, TDIL takes the environment's dynamics into account and encourages the agent to go to the states that are genuinely 'close' (1-step reachable) to the expert states.

- In the single demonstration IL setting, to the best of our knowledge, TDIL is the first algorithm capable of reaching expert-level performance in all the environments we tested. This includes five Mujoco environments (Halfcheetah, Hopper, Walker2d, Ant, and Humanoid) and one adroit manipulation environment (AdroitHandDoor). It is worth noting that in the AdroitHandDoor environment, while two of the best performing baselines (f-IRL, CFIL) exhibit success rate within 40%, the agent trained with our method is able to achieve a 100% success rate.

- The full version of our method that uses $R_{agg} = \beta R_{IRL} + (1 - \beta)R_{TDIL}$ as a reward can train a soft actor-critic (SAC) agent without using BC and still reach expert performance in all the environments we tested. We provide more details in Table A5 of the supplementary material for reviewers’ references.

- Our proposed surrogate reward concept can be utilized to select the best checkpoints without the assistance of the ground truth reward. This is significant since the model that achieves the highest IRL return may differ from the one with the highest ground truth return. Further details are provided in Section 3.3 of the original main manuscript and Section A5 of the supplementary material.

We hope that these clarifications can help the area chairs and the reviewers to better capture the main themes and the key fundamentals of this paper.

Sincerely,

Authors of Paper ID 997.


[r1] Arjovsky, Martin, and Léon Bottou. Towards principled methods for training generative adversarial networks. arXiv:1701.04862 (2017).

---

### Meta-Review · Area_Chair_ELn1 · 2023-12-11

**Metareview:**

This paper is concerned with addressing the challenges of single-demonstration imitation learning (IL), where the learning agent has limited access to expert demonstrations and lacks external rewards or human feedback. The study introduces Transition Discriminator-based IL (TDIL), a methodology that incorporates environmental dynamics and surrogate reward functions to enhance agent performance, mitigate sparse-reward problems, and stabilize the training process. Experimental results across multiple benchmarks validate the effectiveness of TDIL compared to existing IL methods. Some weaknesses of the paper may include the novelty of the idea, the presentation, and the experiment comparison.

**Justification For Why Not Higher Score:**

There are some weaknesses of the paper raised from the review comments and discussions, include the novelty of the idea, the presentation, and the experiment comparison.

**Justification For Why Not Lower Score:**

N/A

---

### Decision · Program_Chairs · 2024-01-16

Reject